# Rapid glacial retreat on the Kamchatka Peninsula during the early 21st Century

Colleen M. Lynch[1], Iestyn D. Barr[1], Donal. Mullan[1], Alastair. Ruffell[1]

[1] School of Geography, Archaeology and Palaeoecology, Queen's Univeristy, Belfast, BT7 1NN, United Kingdom

*Correspondence to*: Colleen M. Lynch (clynch31@qub.ac.uk)

**Abstract.** Monitoring glacier fluctuations provides insights into changing glacial environments and recent climate change. The availability of satellite imagery offers the opportunity to view these changes for remote and inaccessible regions. Gaining an understanding of the ongoing changes in such regions is vital if a complete picture of glacial fluctuations globally is to be established. Here, satellite imagery (Landsat 7, 8 and ASTER) is used to conduct a multi-annual remote sensing survey of

glacier fluctuations on the Kamchatka Peninsula (Eastern Russia) over the 2000–2014 period. Glacier margins were digitised manually, and reveal that in 2000, the peninsula was occupied by 673 glaciers, with a total glacier surface area of $775.7 \pm 27.9$ $km^2$. By 2014, the number of glaciers had increased to 738 (reflecting the fragmentation of larger glaciers), but their surface area had decreased to $592.9 \pm 20.4$ $km^2$. This represents a ~ 24% decline in total glacier surface area between 2000 and 2014, and a notable acceleration in the rate of area-loss since the late 20th century. Analysis of possible controls indicates that these

glacier fluctuations were likely governed by variations in climate (particularly rising summer temperatures), though the response of individual glaciers was modulated by other (non-climatic) factors, principally glacier size, local shading and debris-cover.

## 1 Introduction

Since glaciers are intrinsically linked to climate (Oerlemans et al., 1998), fluctuations in their dimensions are some of the best

natural indicators of recent climate change (Lemke et al., 2007; Paul et al., 2009). In recent years, the improved quality and availability of satellite imagery has allowed fluctuations of glaciers in isolated and often inaccessible regions to be studied remotely (Gao and Liu, 2001; Raup et al., 2007). This can reveal key information concerning the changing local climate, and provide insights into specific controls on glacier behaviour (e.g., allowing the role of climatic forcing and non-climatic modulation to be assessed) (Tennant et al., 2012; Stokes et al., 2013; Burns and Nolin, 2013). Given this utility, we use remote

sensing methods to investigate recent (2000–2014) fluctuations in the surface area of glaciers on the Kamchatka Peninsula (Eastern Russia), and consider possible climatic and non-climatic impacts on this behaviour. The Kamchatka Peninsula is of particular interest because investigation of its recent glacial history has been limited (c.f., Khromova et al., 2014; Earl and Gardner, 2016), despite being the largest glacierised area in NE Asia (Solomina et al., 2007), and a region where glaciers, climate and active volcanoes currently interact (Barr and Solomina, 2014).

## 2 Study Area

### 2.1 Topography

Located in Far Eastern Russia, the Kamchatka Peninsula occupies ~ 260,000 km$^2$. It extends approximately 1,250 km from
the Koryak Highlands in the north to Cape Lopatka in the south, and separates the North Pacific Ocean to the east from the
Sea of Okhotsk to the west (Fig. 1). The peninsula has been shaped by both volcanic and glacial forces (Braitseva et al., 1995,
1997; Ponomareva et al., 2007, 2013; Barr and Clark, 2012 a, b), and is dominated by three mountain ranges: the Sredinny
(Central) Range, the Eastern Volcanic Plateau and the Vostocny (Eastern) Range (Fig. 1). Between these ranges lies the Central
Kamchatka Depression, with relatively flat terrain punctuated by high volcanic peaks (> 4000 m above sea level; a.s.l.). The
three principal mountain ranges are orientated North East to South West, reflecting the position, and eastward movement of
the Kuril-Kamchatka subduction zone, which forms an offshore trench running almost parallel to the peninsula's eastern
coastline (Bulin, 1978). This subduction zone is responsible for the 30 active, and ~ 300 inactive, volcanoes on the peninsula
(Solomina et al., 2007).

### 2.2 Climate

Kamchatka lies between 50°N and 61°N, and 155°E and 163°E, and its proximity to the Pacific Ocean and Sea of Okhotsk
results in a significantly milder climate than much of adjacent Siberia. Average summer temperatures range from 10°C to 15°C
(Ivanov, 2002), whilst average winter temperatures range from -8°C to -10°C in the south east, and from -26°C to -28°C in the
interior and north west, where cold temperatures are amplified by the effects of continentality and higher altitudes. A similar
gradient is visible in the precipitation regime, with average annual values of ~ 1500–2000 mm in the south east and 400–600
mm in the North West. This gradient is partly a consequence of the mountain chains acting as orographic barriers, impeding
and disrupting air-flow from the Pacific (Barr and Clark, 2011). During winter, cold, dry air is drawn from the Siberian High
over central Siberia (Barr and Solomina, 2014). However, this is tempered by the development of a low pressure system over
the Sea of Okhotsk, which sustains precipitation and elevates temperatures during winter (Velichko and Spasskaya, 2002).
During summer, the Pacific High, to the southeast, brings warm-moist air masses across the peninsula. Precipitation peaks
during autumn, aided by the delayed onset of the East Asian Monsoon (Velichko and Spasskaya, 2002).

### 2.3 Glaciers

At present, glaciers on the peninsula are found within the three principal mountain ranges, and on the high volcanic peaks of
the Central Kamchatka Depression (see Fig. 1). The first attempts to estimate the extent of the region's glaciers were published
as part of the 'Catalogue of Glaciers in the USSR, volume 20' (Vinogradov, 1968; Khromova et al., 2014). This was based on

observations from topographic maps and aerial imagery captured in the early 1950s (various years), supplemented by field studies (Kotlyakov, 1980). In this survey, 405 glaciers were documented, with a total area of ~ 874 km$^2$ (Vinogradov, 1968). This was used as a basis for the USSR Glacier Inventory (UGI), and the dataset was subsequently incorporated within both the World Glacier Inventory (WGI) (WGMS, 1989) and Randolph Glacier Inventory (RGI) version 2 (Arendt et al., 2012).

Subsequent estimates of the total number and surface area of glaciers on the peninsula tend to differ. For example, Muraviev (1999) reports 448 glaciers, with a total area of ~ 906 km$^2$, while Solomina et al. (2007) report 446 glaciers, with a total area of ~ 900 km$^2$. Most recently, Earl and Gardner (2016) conducted extensive mapping of glaciers in Northern Asia using automated remote sensing techniques (i.e. NDSI), with a size threshold of 0.02 km$^2$, and documented 984 glaciers on Kamchatka, covering a total area of 770.3 km$^2$. This data was incorporated into the most recent version of the RGI (version 5,
released in July 2015) (Arendt et al., 2015). However, it is of note that the Earl and Gardner (2016) dataset is based on a mosaic of satellite images from multiple years (2000, 2002, 2009, 2011, 2013), meaning their estimate does not reflect glacier extent at a single point in time (or during a single year).

In addition to these peninsula-wide inventories, investigations into recent glacier fluctuations in specific regions of Kamchatka have been conducted (locations identified in Fig. 1). For example, based on the analysis of satellite imagery, data from the
UGI, and aerial photographs, Muraviev and Nosenko (2013) identified a 16.6% decline in glacier surface area in the northern sector of the Sredinny Range between 1950 and 2013. Muraviev (2014), using these same techniques, identified a 19.2% decrease in glacier surface area in the Alney-Chashakondzha region between 1950 and 2013 and a 22.9% decrease on the Kronotsky Peninsula between 1950 and 2010. On the basis of fieldwork (2007–2010) and the analysis of aerial photographs (from 1974), Manevich et al. (2015) documented fluctuations of 27 glaciers on the slopes of the Avachinskaya Volcanic Group
between 1974 and 2010, and found that, during this period, seven glaciers advanced, two retreated, and eight remained largely stationary. Other, smaller scale studies include investigations of glaciers on the slopes of Ichinsky Volcano (55.690°N, 157.726°E) (see Matoba et al., 2007), Koryak Volcano (53.321°N, 158.706°E) (see Manevich and Samoilenko, 2012), in the Klyuchevskaya Volcanic group (~ 56.069°N, 160.467°E) (see Shiraiwa et al, 2001), and at Koryto glacier (54.846°N, 161.758°E) on the Kronotsky Peninsula (see Yamaguchi et al., 1998, 2007, 2008). As a result of these investigations, some of
the glaciers in the Central Kamchatka Depression have been documented as 'surge-type' (Vinogradov et al., 1985). For example, Bilchenok glacier (56.100°N, 160.482°E) in the Klyuchevskaya volcanic group, is known to have surged ~ 2 km in 1959/1960 and again in 1982/84 (Muraviev et al., 2012). In this example, surging appears unrelated to climate, and is likely driven by the strengthening of seismic activity at Ushkovsky Volcano (56.069°N, 160.467°E), upon which the glacier sits (Muraviev et al., 2012). Direct mass balance observations from Kamchatka's glaciers are limited, with the longest continuous
record obtained for Kozelsky glacier (53.245°N, 158.846°E), spanning the 1973–1997 period. Other mass balance measurements, from Koryto, Mutnovsky SW (52.448°N, 158.181°E), Mutnovsky NE (52.460°N, 158.220°E), and Kropotkina (54.321°N, 160.034°E) glaciers, are far shorter and often discontinuous (see Barr and Solomina, 2014).

## 3 Methods

### 3.1 Data sources

Changes in glacier extent on the Kamchatka Peninsula between 2000 and 2014 were determined through visual analysis of multi-spectral Landsat 7 Enhanced Thematic Mapper Plus (ETM+), Landsat 8 Operational Land Imager (OLI), and Terra

Advanced Spaceborne Thermal Emission and Reflection Radiometer (ASTER) satellite images (see Table 1). Orthorectified scenes were obtained from the USGS Earth Explorer website (http://earthexplorer.usgs.gov/) and projected using WGS 1984 Universal Transverse Mercator (UTM) Zone 57N, then filtered to find those with minimal cloud- and snow-cover. To achieve this, images were restricted to those captured during the ablation season (for Kamchatka this occurs from late July to early September). Satellite images captured prior to 2000 (e.g., Landsat MSS and CORONA) were obtained and analysed, however

images captured during the ablation season, with minimal snow- and cloud-cover were limited, and full coverage of the peninsula's glaciers was lacking. As a result, the present study only focuses on 2000 and 2014 (i.e., those years with full satellite coverage of the region's glaciers. In total, 14 Landsat 7 (ETM+) scenes and one ASTER image were used for 2000, with 14 Landsat 8 (OLI) scenes used for 2014 (see Table 1). The Landsat scenes have a spatial resolution of 30 m for all bands except panchromatic band 8 (15 m) and Thermal Infrared Sensor (TIRS) band 6 (60 m), whilst the ASTER image (bands 1–

3) has a spatial resolution of 15 m. To analyse glacier topography, SRTM 1 Arc-Second Global elevation data was obtained from the USGS Earth Explorer website. This reflects the surface topography of the glaciers in February 2000, with a horizontal resolution of ~ 30 m, and with vertical errors < 16m (at the 90% confidence level) (Farr et al., 2007).

### 3.2 Delineation of glacier margins

As highlighted by Paul et al. (2013), the identification and mapping of glaciers from satellite images can be conducted using

semi-automated techniques (e.g. Paul and Andreassen, 2009; Kamp and Pan, 2015; Smith et al., 2015; Earl and Gardner, 2016) or through manual digitising (e.g. DeBeer and Sharp, 2007; Stokes et al., 2013). In the present study, in order to establish the best method for mapping glaciers on Kamchatka, both semi-automated techniques and manual digitisation were trialled. The semi-automated techniques (specifically a RED/SWIR ratio and Normalised Difference Snow Index (NDSI)) produced rapid outputs and were easily applied across large regions, but had difficulty differentiating ice from the surrounding environment

(Fig. 2). The RED/SWIR band ratio had particular problems in distinguishing shaded areas and dirty ice, while the NDSI sometimes failed to distinguish ice from clouds and water bodies. Due to these difficulties, combined with the extensive manual editing and post-processing required to produce precise glacier margins, the semi-automated techniques were rejected in favour of manual mapping. Although time-consuming and sometimes subjective (Bhambri et al., 2011; Paul et al., 2013; Pfeffer et al., 2014), manual mapping remains the best method to extract detailed information on glacier dimensions (Kääb, 2005), whilst

additionally allowing snow patches and areas covered by shadow to be identified (see Fig. 2).

To help differentiate glaciers from their surroundings and aid in the visual detection of glacier margins, both true- and false-colour composites of the satellite images were generated (Malinverni et al., 2008), with the high resolution band 8 (15 m) used

to sharpen the image. The true-colour composites were used for the identification of glaciers, while the false-colour composites helped differentiate ice and cloud, due to the strong absorption of ice in the SWIR band when compared to clouds (Nuimura et al., 2015). Glaciers were mapped in ArcGIS 10.2.2 (by a single operator) as polygon shapefiles, and features $< 0.02$ km$^2$ in 2000 were excluded from the analysis on the grounds that they are likely to be snow- or ice-patches, rather than glaciers (Paul and Andreassen, 2009; Paul et al., 2009; Bajracharya and Shresta, 2011; Frey et al., 2012; Racoviteanu et al., 2015). To help verify that each of the digitised polygons was a glacier rather than a transient snow-patch, scenes from intervening years, including 2015, were also obtained and analysed (Paul et al., 2009). However, the glacier inventory produced here does not contain detailed information about glacier dimensions for these intervening years since complete peninsula-wide imagery without extensive snow- or cloud-cover was unavailable for these periods, meaning that not all glaciers were mapped. For each of the glaciers mapped in 2000 and 2014, two-dimensional surface area was measured directly from the shapefiles, and area differences between 2000 and 2014 were calculated. Where glaciers fragmented into several distinct ice masses over this period, the net change in area was calculated based on the sum of the total area of each of the glacier 'fragments' (following DeBeer and Sharp, 2007). For each glacier identified in the inventory in 2000, maximum, minimum and median altitude and mean surface slope were calculated from the SRTM DEM, and generalised glacier aspect was estimated from a line connecting the glacier's maximum and minimum altitudes. The mean annual receipt of solar radiation at the surface of each glacier was calculated using the Solar Radiation tool in ArcGIS (algorithms developed by Fu and Rich, 2002), and glacier length was estimated along inferred flowlines.

### 3.3 Error Estimation

Potential sources of error in this study arise through the digitisation process and with difficulties in correctly identifying areas of glacial ice. The accuracy of digitisation depends partly on the spatial resolution of the satellite imagery, snow conditions and the contrast between glacier ice and the surrounding environment (DeBeer and Sharp, 2007; Stokes et al., 2013). In Kamchatka, it is notable that the maritime conditions create particular difficulty with locating cloud-free imagery, and persistent snow-cover often hinders the clear identification of glacier margins (Paul and Andreassen, 2009). Where glacier margins were partially obscured by shadow, debris or cloud, careful analysis using multiple true- and false-colour composite band images, as well as thermal imagery, was undertaken. Error was calculated following the method described in Bajracharya et al. (2014):

$$RMSE = \sqrt{\frac{\sum_{i=1}^{n}(a_i - â_i)^2}{n}} \tag{1}$$

where $a_i$ denotes glacier area and $â_i$ is the glacier area calculated on the pixel base (i.e., the total number of pixels within a polygon, multiplied by the highest image resolution (15 m) used for mapping), and n is the number of polygons digitised. Error was found to be ~ 3.6% in 2000 and ~ 3.4% in 2014. These values are comparable to those reported in other studies (Bolch et al., 2010; Bhambri et al., 2013; Bajracharya et al., 2014).

### 3.4 Climate Data

NCEP/NCAR reanalysis data (see Kalnay et al., 1996), was used to analyse recent fluctuations in climate across Kamchatka. The NCEP reanalysis data integrates available land surface data from climate stations with the recovery of data from radiosondes, aircraft, ships and satellites to assimilate a quality controlled gridded (1.88° latitude x 1.88° longitude) record of surface climate variables for the period 1948 – present (Kalnay et al., 1996). The precipitation data reflect surface conditions, and temperature data reflect conditions at a height of 2 m (a.s.l). Reanalysis data was used in preference to analysing individual weather stations, since the latter are often situated in coastal locations, distant from Kamchatka's glaciers. Additionally, many of the available station records include periods of missing data.

### 4 Results

#### 4.1 Glacier inventory in 2000

In total, 673 glaciers, with a combined surface area of $775.7 \pm 27.9$ km$^2$, were identified and mapped on the peninsula in 2000 (see Fig. 1). Summary statistics for these glaciers are presented in Table 2, revealing a mean surface area of 1.15 km$^2$ (ranging from 0.02 to 48.82 km$^2$), with ~ 77% < 1 km$^2$, and less than 4% > 5 km$^2$, though the latter represent just over one third of the total glacier area. Glaciers range in length from 0.17 km to 20.96 km, while ranging in altitude from 273 m to 4407 m (.a.s.l.), with a mean altitudinal range of 465 m, and a mean surface slope of 17°. The mean annual receipt of solar radiation at the surface of each glacier ranges from 489.8 to 1012.4 (kWh/m$^2$). Across the peninsula, 52 glaciers have tongues covered with debris (rock and ash), the majority of which exist on volcanic slopes in the Central Kamchatka Depression (see Yamaguchi et al., 2007).

#### 4.2 Changes in glacier number and extent between 2000 and 2014

In 2014, 738 glaciers, covering an area of $592.9 \pm 20.4$ km$^2$, were identified and mapped on the peninsula. This represents an additional 65 glaciers, but an overall area loss of $182.9 \pm 6.6$ km$^2$ (~ 24%), relative to 2000. The increase in glacier number occurred despite the loss of 46 glaciers during this period (these glaciers were all < 0.5 km$^2$ in 2000 and either completely disappeared or fell below the size threshold of 0.02 km$^2$, by 2014) (see Fig. 3), and primarily reflects the fragmentation of larger glaciers. Of the peninsula's 673 glaciers identified in 2000, 654 (~ 97%) experienced a reduction in surface area by 2014, and this decline is seen across the peninsula (Fig. 4). Of the 17 glaciers that increased in area during this period, the majority experienced minor growth. One glacier increased by ~ 140% (see Fig. 4B), but was very close to the size threshold of 0.02 km$^2$ in 2000, and the area gain over the period of observation only equates to 0.03 km$^2$. In addition to areal decline, signs of stationary thinning (downwasting) are apparent in many areas, reflected by the exposure of many rock outcrops and the fragmentation of glaciers (see Fig. 5). The exposure of bedrock in this way further accelerates glacier wastage as local albedo is reduced (Paul et al., 2007).

## 4.3 Climate

When analysed across the peninsula, the NCEP data appear to indicate a reduction in precipitation between the 1970s and 1990s, but a precipitation increase thereafter (see Fig. 6A). This recent increase in precipitation is particularly notable during autumn (Fig. 6A), though more importantly for glacier mass balance, winter precipitation also increases during this period, likely contributing directly to increased snow accumulation. The temperature data (Fig. 6B) appears to show a warming trend from the 1950s to the late 1990s, which has continued (with some fluctuations) to the present day. Most significantly, there has been a sharp increase in average summer temperatures (June, July and August) since the early 21$^{st}$ century (see Fig. 6B).

## 4.4 Glacier attributes and glacier area change

Glacier size (area, perimeter, length and altitudinal range), altitude (maximum, median and minimum), surface slope and aspect (based on the glaciers mapped in 2000) were compared to absolute and relative changes in glacier surface area across the peninsula. These comparisons reveal a strong negative, and statistically significant ($p < 0.001$) correlation between the 2000 to 2014 change in glacier surface area (in km$^2$) and glacier area ($r = -0.75$), length ($r = -0.65$) and altitudinal range ($r = -0.48$) in 2000 (Fig. 7A–C). When change in area is expressed as a percentage of the original glacier size (rather than total area), these correlations become positive ($r = 0.24$, 0.34 and 0.34 for surface area, length and altitudinal range, respectively), and are weakened (though they remain statistically significant) (Fig. 7D–F). There is also a negative, and statistically significant, correlation between glacier surface area change (in km$^2$) and the maximum ($r = -0.37$) and median ($r = -0.25$) glacier altitude. Again, when area change is expressed as a percentage, these correlations become positive ($r = 0.23$ and 0.17 for maximum and median, respectively), and are weakened (though they remain statistically significant). When glacier minimum altitude or surface slope are considered, there is no statistically significant relationship with glacier area loss (when expressed as total area, or as percentage change). Glaciers are predominantly found with an aspect-bias towards northerly and western directions (54.98 %), and glacier aspect shows a weak ($r = 0.15$), but statistically significant, relationship with changes in glacier surface area, though only when expressed in km$^2$, rather than as a percentage. Similarly, analysis of insolation patterns reveals a weak, but statistically significant correlation ($r = 0.18$) with change in glacier surface area, but only when expressed in km$^2$, rather than as a percentage.

Based on mapping in 2000, ~ 8% (n = 52) of the peninsula's glaciers are classed as debris-covered. When the glacier population is split into 'debris-covered' and 'non debris-covered' samples in this way, the former lose on average ~ 11% (0.62 km$^2$) of their surface area between 2000 and 2014, while the latter lose ~ 44% (0.24 km$^2$). Thus, over the 2000–2014 period, debris-covered glaciers lose less surface area than non debris-covered examples, though this relationship is only statistically significant when values are expressed as a percentage, rather than in km$^2$.

# 5 Discussion

## 5.1 Comparison with other inventories

Based on the data presented in the Catalogue of Glaciers in the USSR (see Sect. 2.3.), and the findings of the present study, there was a ~ 11.2% decline in glacier surface area on the Kamchatka Peninsula between the 1950s (exact date unspecified) and 2000, and a further ~ 23.6% loss between 2000 and 2014. Assuming linear trends, this indicates an area-loss rate of ~ 0.24–0.29 % $a^{-1}$ between 1950 and 2000, and a notable acceleration to ~ 1.76% $a^{-1}$ since 2000. The reduction in glacier area during the late 20th century coincides with negative trends in glacier mass balance on the peninsula (see Barr and Solomina, 2014). Unfortunately, since 2000, mass balance data have not been collected for Kamchatka's glaciers, and it is not possible to assess whether accelerated mass loss over the early 21$^{st}$ century has coincided with the accelerated glacier shrinkage identified here.

Similar rates of retreat have been found for glaciers in other Asian mountain ranges. For example,  in the Altai Mountains, Narozhniy and Zemtsov (2011) document a 10.2% decrease in glacier area between 1956 and 2008, while Kamp and Pan (2015) document a 13% decrease between 1998/2001 and 2010/2011. In the Tien Shan Mountains, Farinotti et al. (2015) document an $18 \pm 6\%$ decrease in glacier surface area between 1961 and 2012. In the Kodar Mountains, Stokes et al. (2013) document a 44% decrease in the area of exposed glacial ice between 1963 and 2010, with a 40% loss since 1995, and in the Caucasus Mountains, Tielidze (2016) documents a $36.9 \pm 2.2\%$ decline in glacier surface area between 1960 and 2014, though regional differences are noted.

In terms of glacier surface area, it is notable that our estimate from 2014 ($592.9 \pm 20.4$ km$^2$) differs significantly from Earl and Gardner's (2016) estimate (770.3 km$^2$). This we attribute to their semi-automated approach to mapping (i.e., NDSI), which can lead to an overestimation of glacier area as a result of snow patches being erroneously classified as glaciers (Man et al., 2014) (see Fig. 8), combined with the fact that their inventory was generated from a composite of satellite images, meaning their mapping does not reflect glacier extent during a single specified year (as noted in Sect. 2.3.).

## 5.2 Potential climatic controls on glacier fluctuations

In the small number of studies to consider the issue (noted in sect. 2.3), the retreat of Kamchatka's glaciers over recent decades has typically been attributed to variations in climate. For example, Yamaguchi et al. (2008) consider the retreat of Koryto glacier between 1711 and 2000 to be a result of decreased precipitation over this period. Similarly, in the Northern part of the Sredinny Range, Muraviev and Nosenko (2013) note that from 1950 to 2002, average summer temperatures increased, while solid precipitation (snowfall) decreased, and suggest that this caused the retreat of the region's glaciers.

Based on climatic trends identified from the NCEP data, it might be argued that the loss of glacier surface area across Kamchatka between the 1950s and 2000 reflects the combined influence of rising temperatures and declining precipitation. However, since 2000, despite an increase in precipitation, glacier area loss has continued, and appears to have accelerated.

This might reflect a delayed response to earlier, drier, conditions, but is also likely driven by a notable increase in temperatures since the mid-1990s. In particular, the pronounced rise in summer temperature (see Fig. 6C) is likely to have increased the intensity of melt, whilst rising autumn temperatures (Fig. 6B) may have lengthened the ablation season (simultaneously shortening the accumulation season). Therefore, it would appear that temperature has been the primary control on early 21[st] Century glacier fluctuations in Kamchatka, with rising temperatures driving a notable decline in glacier surface area, despite a corresponding rise in precipitation (even during winter).

### 5.3 Potential non-climatic modulation of glacier fluctuations

Despite evidence for climatic control over glacier fluctuations in Kamchatka (outlined in Sect. 5.2.), the individual response of these glaciers is likely to have been modulated by other, local, non-climatic factors (Evans, 2006; Tennant et al., 2012; Stokes et al., 2013). These factors potentially include glacier size (area, perimeter length and altitudinal range), altitude surface slope, and aspect, as well as volcanic controls and debris cover (Huss, 2012; Fischer et al., 2015).

### 5.3.1 Glacier size

Comparisons between glacier size and surface area fluctuations suggest that smaller glaciers, though losing least surface area, actually lost a greater proportion of their total area. Similar trends, with small glaciers showing a propensity to shrink rapidly, have been found in numerous regions globally (see Ramírez et al., 2001; Granshaw and Fountain, 2006; Racoviteanu et al., 2015; Tennant et al., 2012; Stokes et al., 2013). This is considered a result of the greater volume-to-area and perimeter-to-area ratios of smaller glaciers—meaning they respond rapidly to a given ablation rate (Granshaw and Fountain, 2006; Tennant et al., 2012). This rapid decline in the area of smaller glaciers on the Kamchatka Peninsula could result in the loss of many over coming decades, as ~ 75% of the glaciers mapped in 2014 have an area < 0.5 km$^2$, of which ~ 87 % have a maximum altitude < 2000 m (a.s.l.), likely making them particularly sensitive to future warming. This supports the view of Ananicheva et al. (2010), who suggest that by 2100, only the largest glaciers on the highest volcanic peaks will remain.

### 5.3.2 Glacier altitude

Comparisons between glacier median and maximum altitude and surface area fluctuations would appear to suggest that these factors exert some control on area change. However, the lack of any statistically significant relationship between area loss and minimum altitude might indicate that, rather than exerting a direct control on glacier area, glaciers with high maximum and median altitudes are typically the largest on the peninsula (i.e., there are positive and statistically significant relationships between glacier area and both maximum ($r = 0.37$) and median ($r = 0.25$) altitude), and that size exerts the primary control on glacier behaviour in this relationship. Similar trends, highlighting the relationship between glacier area and elevation, have been observed in other regions globally, for example in the Kanchenjunga–Sikkim region of the Himalayas (Racoviteanu et al., 2015) and in Cordillera Blanca, Peru (Racoviteanu et al., 2008).

### 5.3.3 Glacier aspect

The aspect bias exhibited by Kamchatkan glaciers, combined with the statistically significant relationship between area loss and total insolation indicates that glaciers exposed to most solar radiation typically show a greater reduction in their overall surface area (see Evans, 2006). However, though statistically significant, the relationship between glacier aspect and changes in glacier surface area is comparatively weak, suggesting that local variations in insolation (e.g., related to topographic shading) are likely important in protecting glaciers from recession (see Paul and Haeberli, 2008; Stokes et al., 2013)

### 5.3.4 Volcanic controls and debris cover

Across the peninsula, 165 glaciers are located within a 50 km radius of an active volcano, with 292 glaciers within a 100 km radius. During the period of observation (i.e., 2000–2014), > 40 individual volcanic eruptions were documented on the peninsula (VONA/KVERT, 2016). Although, there is evidence that some glaciers were covered by tephra as a result of these eruptions, there is no evidence to suggest that this had a discernible influence on glacier fluctuations during the period of observation. This likely reflects the longer response time of glaciers to tephra deposition, since a number of Kamchatka's glaciers are known to have responded to volcanic ash cover over longer time periods (see Barr and Solomina, 2014). It is also apparent, from the present study, that debris-covered glaciers on the peninsula are typically less responsive to external forcing, since they lose less surface area (both in absolute and, particularly, relative terms) than non debris-covered examples. Similar patterns have been observed for debris covered glaciers elsewhere. For example, in the Kanchenjunga–Sikkim area, largely debris-free glaciers showed a 34% decline in surface area between 1962 and 2006, whilst debris covered glaciers experienced a 22% decline over this period (Racoviteanu et al., 2015). This tendency for debris-covered glaciers to retreat comparatively little might reflect the insulating influence of accumulated surface debris (Vinogradov et al., 1985), or might indicate that the terminus positions of such glaciers are inclined to stabilise even when the glaciers are experiencing mass loss. Such a trend was identified in the Central Tien Shan, where Pieczonka and Bolch (2015) found that glaciers (many of which are debris-covered) experienced comparatively little area change (below the global average) between 1975 and 1999, despite mass loss similar to the global average over this period.

In addition to the influence exerted by surface debris, some of Kamchatka's glaciers are known to be of 'surge-type', with surface indicators of past surge activity (e.g., looped moraines and heavy crevassing) (see Copland et al., 2003), and with documented surges during the 20[th] Century (as noted sect. 2.3) (Vinogradov et al., 1985; Yamaguchi et al., 2007). However, there was no evidence of surging during the period of observation. This might reflect the comparatively short time period considered, or may be a reflection of volcanically controlled surging (see section 2.3) (Muraviev et al., 2012), which therefore lacks periodicity.

**6 Conclusion**

In this paper, manual digitisation from satellite imagery is used to map the surface area of all glaciers on the Kamchatka Peninsula in 2000 and 2014. This is the first study to consider peninsula-wide patterns in glacier behaviour over the early 21$^{st}$ Century, and variations in glacier extent are put in context through comparison with published glacier extent estimates from the 1950s (Kotlyakov, 1980). The main study findings can be summarised as follows:

1. In total, 673 glaciers, with a combined surface area of $775.7 \pm 27.9$ km$^2$, were identified and mapped on the peninsula in 2000. By 2014, the total number of glaciers had increased to 738 but their surface area had reduced to $592.9 \pm 20.4$ km$^2$, this suggests an acceleration in the rate of area loss since 2000 (from $\sim 0.24$–$0.29$ % a$^{-1}$ between the 1950s and 2000, to $\sim 1.76\%$ a$^{-1}$, between 2000 and 2014). The increase in glacier number, despite the disappearance of 46, is considered to reflect the fragmentation of larger glaciers during this period.

2. Based on the analysis of NCEP/NCAR reanalysis climate data, it appears that the reduction in glacier surface area on the peninsula between the 1950s and 2000 likely reflects the combined influence of rising temperatures, and declining precipitation. However, accelerated area loss since 2000, despite increased precipitation, is likely a response to a notable increase in temperatures across the peninsula since the 1990s. Specifically, the rise in summer temperatures is likely to have enhanced the intensity of melt, whilst rising autumn temperatures may have lengthened the ablation season, simultaneously shortening the accumulation season.

3. Despite the overall climatic control there is evidence that the behaviour of individual glaciers on the peninsula is modulated by local, non-climatic factors. Specifically, smaller glaciers, though losing least absolute surface area, lost a greater proportion of their total area. This propensity to shrink rapidly is considered to reflect the greater volume-to-area and perimeter-to-area ratios of smaller glaciers, meaning that they have a heightened sensitivity to changing climate (see Granshaw and Fountain, 2006). Though glacier altitude shows some relation with area change, this probably reflects the positive relationship between glacier altitude and size (rather than an altitudinal control on glacier behaviour). Insolation patterns show a weak, but statistically significant, relationship with changing glacier surface area, indicating that glaciers exposed to most solar radiation experienced a greater reduction in their overall surface area. However, though statistically significant, the relationship between glacier aspect and changes in glacier surface area is comparatively weak, suggesting that local variations in insolation (e.g., related to topographic shading) are important in regulating fluctuations of Kamchatka's glaciers.

4. If the rapid decline in the surface area of smaller glaciers on the Kamchatka Peninsula continues over the 21$^{st}$ Century, many will be lost by 2100 (Ananicheva et al., 2010), since $\sim 75\%$ of the region's glaciers identified in 2014 have an area $< 0.5$ km$^2$, of which $\sim 87\%$ have a maximum altitude $< 2000$ m (a.s.l.), likely making them particularly sensitive to future warming.

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

Table 1. Satellite images used to generate the glacier inventory

| 2000 | | | | 2014 | | | |
|---|---|---|---|---|---|---|---|
| Path | Row | Date | Scene Id | Path | Row | Date | Scene Id |
| 98 | 21 | 22/09/2000 | LE70980212000266EDC01 | 97 | 22 | 29/08/2014 | LC80970222014241LGN00 |
| 98 | 22 | 22/09/2000 | LE70980222000250EDC00 | 98 | 19 | 05/09/2014 | LC80980192014248LGN00 |
| 98 | 22 | 06/09/2000 | LE70980222000250EDC00 | 99 | 19 | 27/08/2014 | LC80990192014239LGN00 |
| 98 | 23 | 06/09/2000 | LE70980232000250EDC00 | 99 | 20 | 27/08/2014 | LC80990202014239LGN00 |
| 99 | 18 | 28/08/2000 | LE70990182000241EDC00 | 99 | 21 | 27/08/2014 | LC80990212014239LGN00 |
| 99 | 22 | 28/08/2000 | LE70990222000241EDC00 | 99 | 21 | 12/09/2014 | LC80990212014255LGN00 |
| 99 | 23 | 28/08/2000 | LE70990232000241EDC00 | 99 | 22 | 12/09/2014 | LC80990222014255LGN00 |
| 100 | 19 | 18/07/2000 | LE71000192000200EDC00 | 99 | 23 | 27/08/2014 | LC80990232014239LGN00 |
| 100 | 20 | 19/08/2000 | LE71000202000232EDC00 | 99 | 23 | 12/09/2014 | LC80990232014255LGN00 |
| 100 | 21 | 19/08/2000 | LE71000212000232EDC00 | 99 | 24 | 12/09/2014 | LC80990242014255LGN00 |
| 100 | 22 | 19/08/2000 | LE71000222000232EDC00 | 100 | 19 | 19/09/2014 | LC81000192014262LGN00 |
| 100 | 23 | 20/09/2000 | LE71000232000264EDC00 | 100 | 20 | 19/09/2014 | LC81000202014262LGN00 |
| 100 | 24 | 20/09/2000 | LE71000242000264EDC00 | 101 | 21 | 10/09/2014 | LC81010212014253LGN00 |
| 99 | 19 | 20/07/2000 | AST_L1T_00307202000004626 _20150410120043_53236 | 101 | 22 | 10/09/2014 | LC81010222014253LGN00 |

Table 2. Statistics for glaciers on the Kamckatka Peninula in 2000 (topographic attributes are derived from the SRTM 30m DEM).

| | Min | Mean | Max |
|---|---|---|---|
| Area (km$^2$) | 0.02 | 1.15 | 48.82 |
| Minimum Altitude (m.a.s.l) | 273 | 1277 | 2920 |
| Median altitude (m.a.s.l.) | 544 | 1506 | 3539 |
| Maximum altitude (m.a.s.l.) | 577 | 1742 | 4407 |
| Altitudinal range (m) | 17 | 465 | 3104 |
| Maximum flowline length (km) | 0.17 | 1.74 | 20.96 |
| Mean surface slope (°) | 5.7 | 17.0 | 35.5 |
| Mean annual solar radiation (kWh/m$^2$). | 489.8 | 817.1 | 1012.4 |

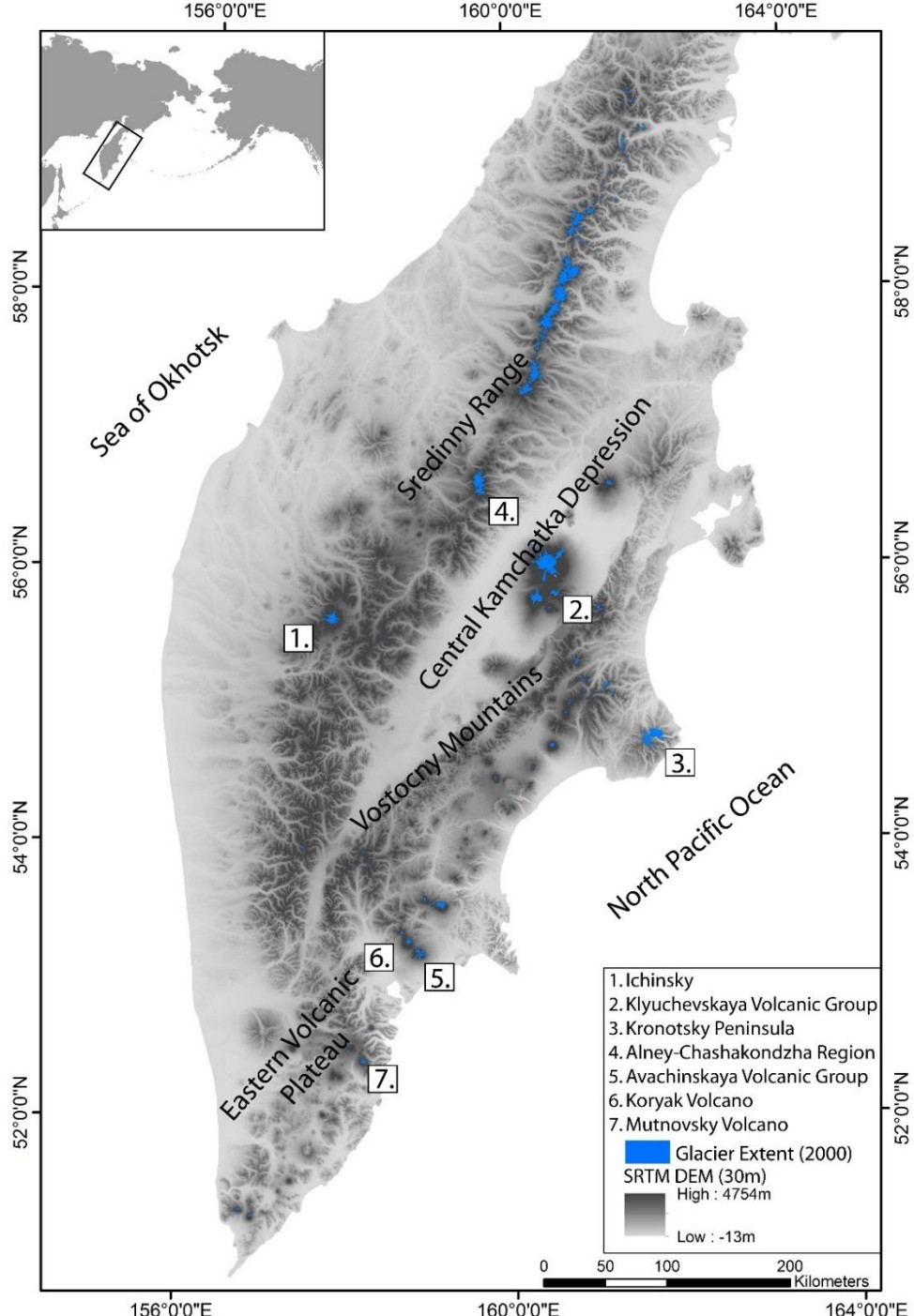

**Figure 1: Map of the Kamchatka Peninsula (Russia) with glacier extent in 2000 shown in blue. Numbers refer to locations discussed in the text.**

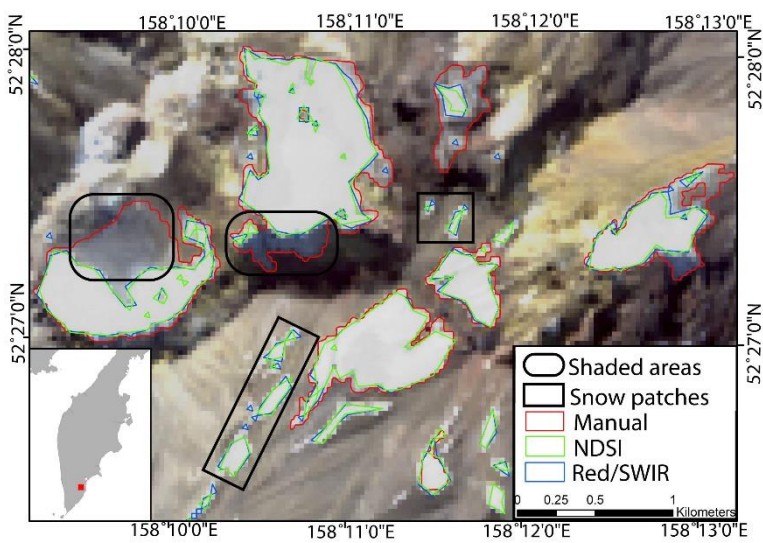

**Figure 2: Comparison of glacier margins (on the slopes of Mutnovsky Volcano) delineated by two semi-automated techniques and manual mapping, (Landsat 2014 background image). This figure illustrates how manual mapping can help identify snow patches and shaded areas of glacier ice, thereby allowing the former to be discounted from, and the latter included in, the glacier inventory.**

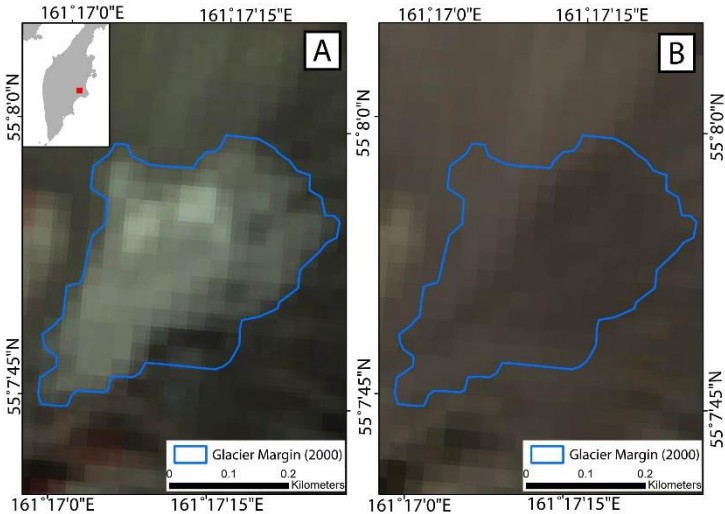

**Figure 3: Example of a small glacier in the Vostocny Mountains that disappeared between 2000 (A) and 2014 (B). In both (A) and (B), the background is a Landsat image.**

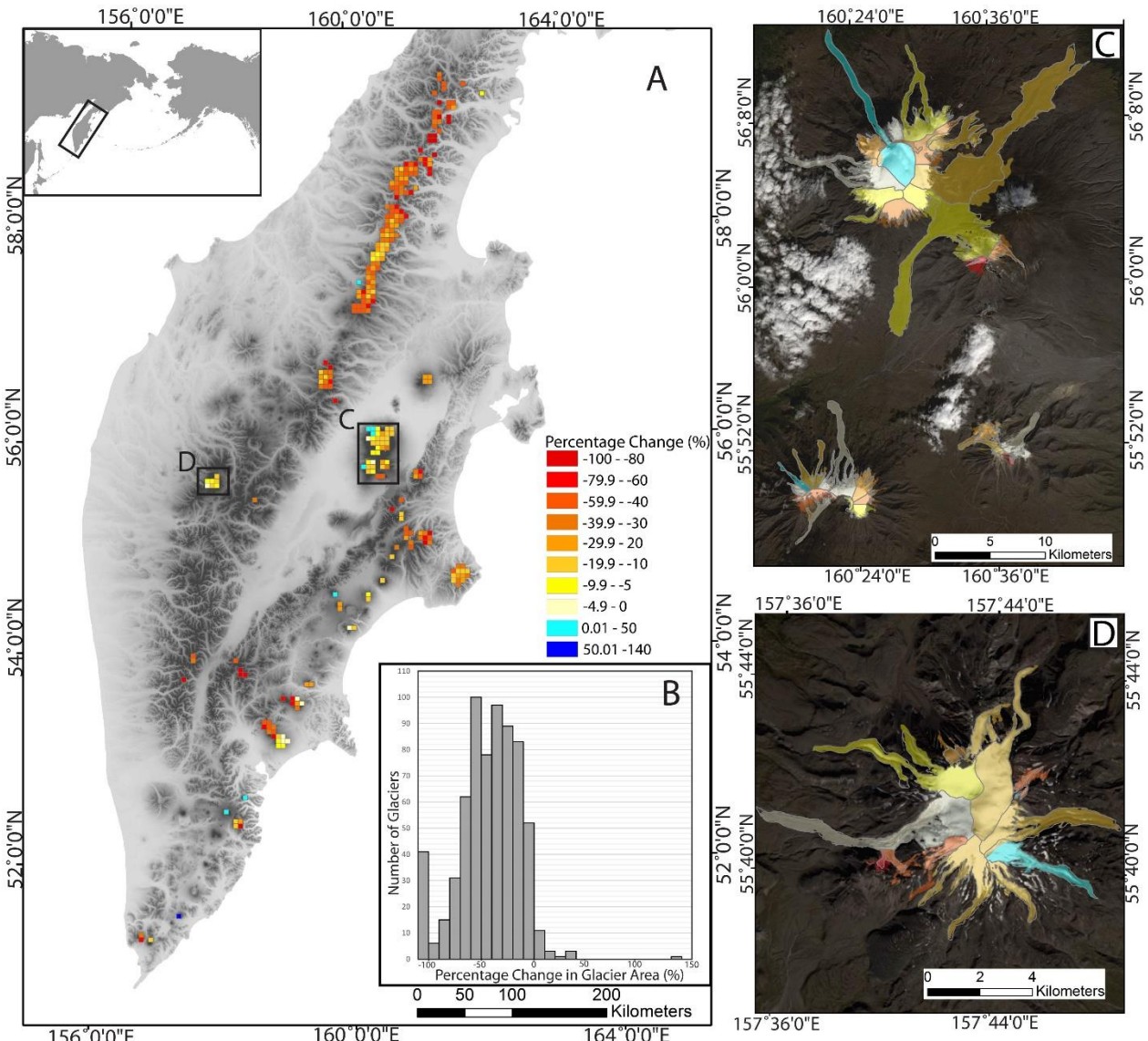

**Figure 4: Percentage change in glacier surface area across the Kamchatka Peninsula between 2000 and 2014. (A) Mean values shown for 5 x 5 km grid cells. (B) Values for individual glaciers shown as a frequency distribution. (C) Values for individual glaciers in the Klyuchevskaya Volcanic Group. (D) Values for individual glaciers on the slopes of Ichinsky Volcano. In both (C) and (D), the colour scheme from (A) is used.**

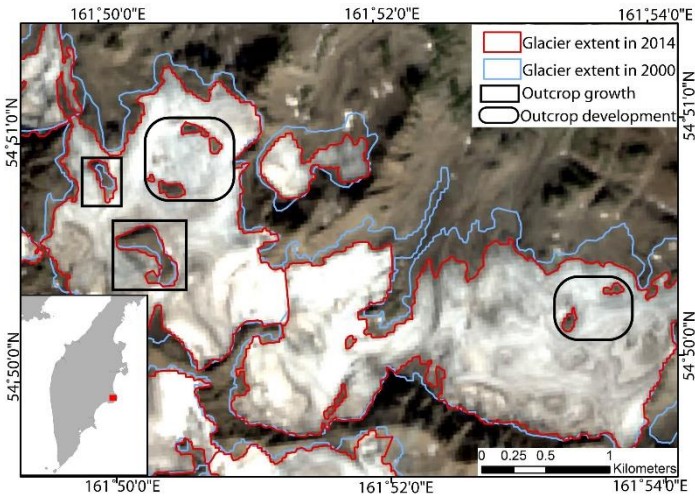

**Figure 5: Examples of glaciers on the Kronotsky Peninsula mapped in 2000 and 2014 (Landsat 2014 background image), revealing stationary thinning and a notable decline in glacier surface area.**

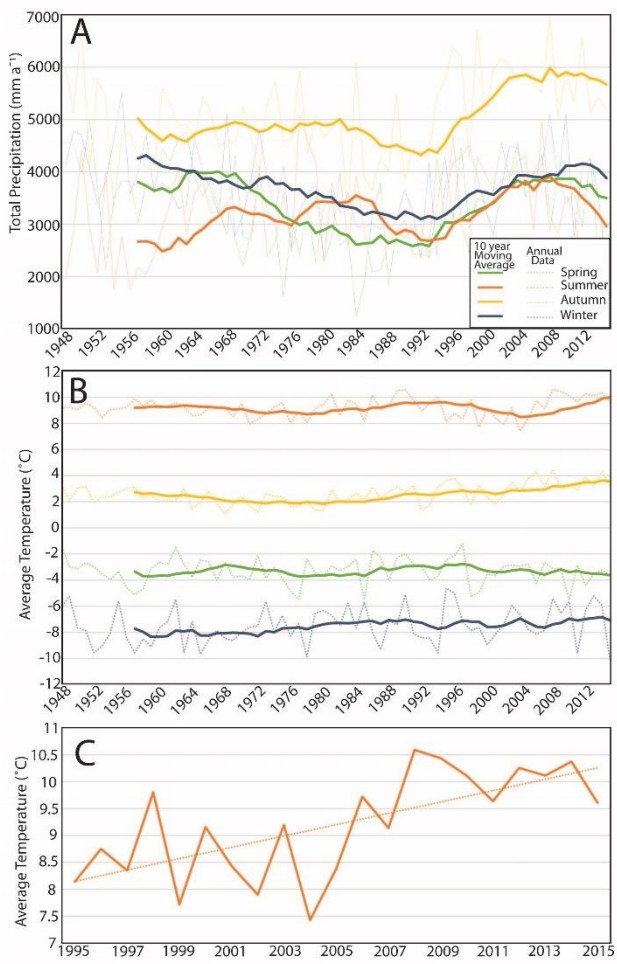

**Figure 6: Climatic Variation on the Kamchatka Peninsula between 1948 and 2015, derived from NCEP/NCAR daily reanalysis figures averaged across the whole peninsula (see Kalnay at al., 1996). (A). Seasonal precipitation record. (B). Average seasonal temperature. (C). Summer temperature record from 1995-2015.**

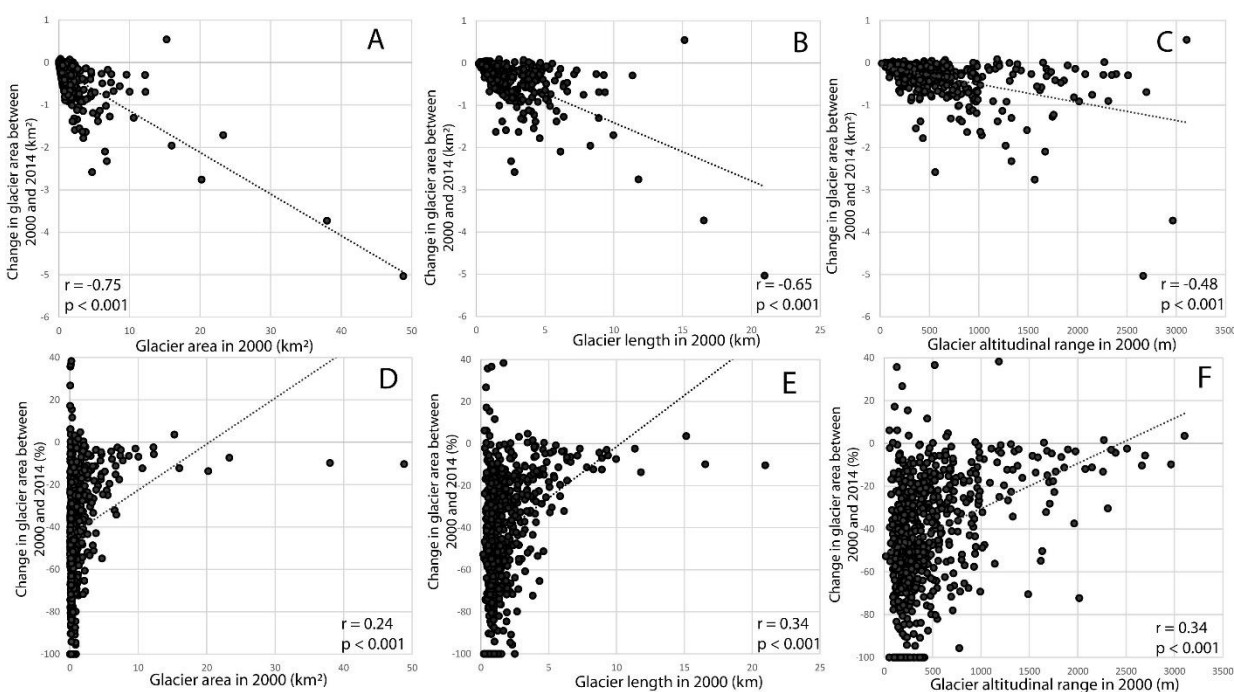

**Figure 7: Relationships between glacier area changes from 2000 to 2014 (in km$^2$) and glacier (A) surface area, (B) length (C) altitudinal range. These same relationships, but with percentage area loss are plotted in (D), (E) and (F).**

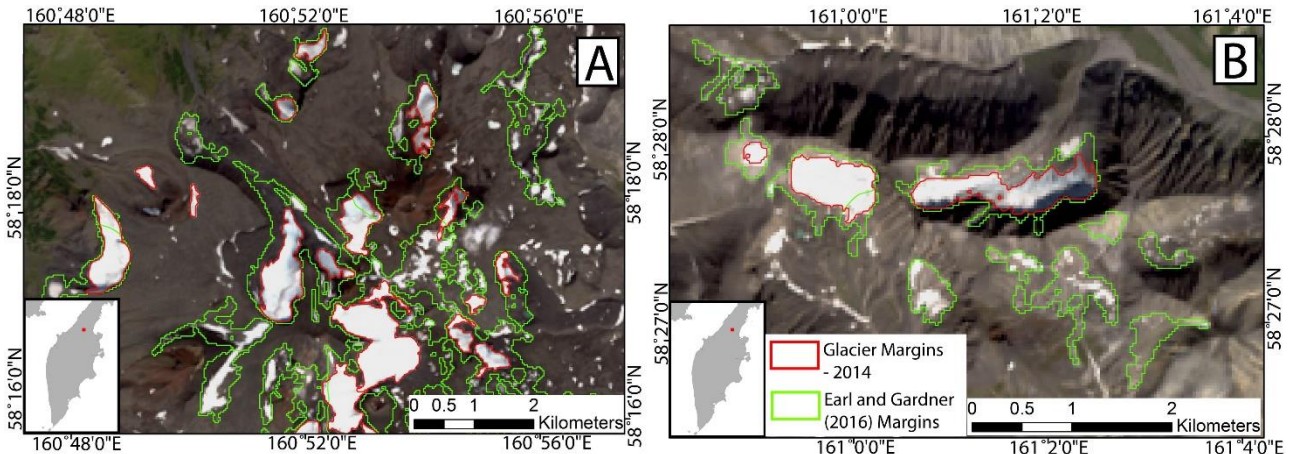

**Figure 8: Comparisons, from across Kamchatka, between our mapping (based on Landsat 8 imagery captured in 2014) and the mapping of Earl and Gardner (2016).**