# Peer review of "Rapid glacial retreat on the Kamchatka Peninsula during the early 21st Century"

_The Cryosphere, 2016_

## Short Comment (SC1) · 8 Apr 2016

According to Figure 2 of the proposed paper, both semi-automatic methods are not suitable (or incorrectly used) for identification of glaciers in the zones of modern volcanic activity, the major part of which is covered with a ground moraine. This person, who manually interpreted the images, has determined not the glacial boundary, but the boundary of surface of open firn and ice. The scheme 1 (see the attached pdf) shows results from determination of glacial boundaries on Tolbachik Volcano based on space images ASTER 19.07.2012, GeoEye-1 01.09.2011 and GeoEye-1 04.07.2013. Data from field observations was used for determination (GPS points and tracks, photos). In order to demonstrate how data of interpretation fit the real setting the images below show glaciers (mainly their terminus) and observation sites. Fine scale diagram in Figure 1 of the paper does not allow adequate assessment of quality of determined

[Figure]

**TCD**

glacial boundary of the volcanic massif with Klyuchevskoy, Kamen, Plosky Blizhny and Plosky Dalny volcanoes as well as Sheveluch Volcano. The authors suppose that the contour shape of the zones occupied by the glaciers in 2000 is enough for conclusion similar to that in paragraph 1 where the authors have determined not the glacial boundary, but the boundary of the surface of open ice and firn. From that follows the conclusion – in this paper, the glaciation area of active Kamchatka volcanism by 2000 is more than twice understated. There is good reason to believe that the situation with the 2014 satellite images interpretation is not better. Ice and firn area size variations on satellite images are caused by many factors, e.g. accumulated seasonal snowfalls, atmospheric temperature during the summer, summer snow flurries etc. It is impossible to determine both the boundary and area of glaciers in general using these area size variations of glacial parts, which are not covered with moraines. The article does not contain the shooting dates for used satellite images. The dates of shooting are important for understanding how the setting of survey corresponds to that in-situ by the end of the ablation period, i.e. the understanding of how the remaining seasonal snow cover and the snow beds could corrupt the results of glacial boundaries determination (especially in the accumulation area). Probably, the errors caused by the remaining seasonal snow cover and the snow beds are the reason for big difference of glaciers areas (as in Figure 4) and their amount between the 2000 and the 2014 images measurements. In comparison with the data of other authors the amount of glaciers found in Kamchatka by authors is underestimated. The authors incorrectly determine the boundaries of the glaciers due to insufficient knowledge of the investigated glaciation area specific features, the lack of the accuracy of the used satellite images and the use of determination methods not valid for glaciers with high debris cover (or misuse of quite valid methods). During the investigation the authors reveal significant reduction of the glaciers area and the increase in their amount from 2000 to 2014, probably explaining by measurement errors resulted from the following factors: the presence of the remaining seasonal snow cover and snow patches at the time of shooting; the lack of the accuracy of the used satellite images Landsat and

[Figure]

ASTER for small (<0.5 km^2) glaciers; incorrect methods of determinations used by the authors. This study does not reflect the changes in glaciation of Kamchatka from 2000 to 2014. The results are based on invalid data.

Please also note the supplement to this comment:
http://www.the-cryosphere-discuss.net/tc-2016-42/tc-2016-42-SC1-supplement.pdf

———————————————————————

[Figure]

**Supplement:**

**Scheme 1**. Tolbachik volcano glaciation. In the substrate ASTER (19.07.2012) satellite image.

[Figure]

[Figure]

**Photo 1**. Institute of Vulcanology Glacier (№ 155) in august 2012.

[Figure]

**Photo 2**. Vinogradova (№ 152) Glacier in august 2013.

[Figure]

**Photo 3**. Kapel'ka Glacier (№ 150) in august 2012.

[Figure]

**Photo 4**. Cheremosnii Glacier (№ 147) in august 2011.

---

## Referee Comment (RC1) · Anonymous Referee #1 · 5 May 2016

This paper presents data on areal changes of comprehensive dataset of glaciers on the Kamchatka Peninsula, obtained using satellite imagery from 2000 and 2014. It reveals a widespread reduction in glacier area, which is compared to older inventory data and to a more recent inventory. The paper then explores possible correlations between glacier shrinkage and climate reanalysis data, in addition to glacier characteristics (e.g. size, elevation, aspect, etc.). The paper is concise (perhaps too concise), well-written and well-organised, with some helpful Figures and Tables (albeit that could be improved and supplemented). It presents an impressive dataset and, given that this region is comparatively poorly studied, I'm generally very supportive of publication in The Cryosphere. However, I would suggest that there are a number of issues that would need to be resolved and clarified prior to publication, which I detail below.

The most important and serious of these issues is that the authors need to be much

[Figure]

clearer about what they have actually mapped and how this compares to other approaches and previous inventories. The authors appear to have mapped "exposed ice", but then make very little comment on how much debris might be obscuring glaciers in this region. In addition to some additional clarification and discussion, it is important that the authors provide some detailed comparisons of their mapping with other approaches. As it stands, the methods and results are somewhat lacking in transparency.

ABSTRACT

P1, line 9: consider deleting "With this in mind"

P1, line 10-12: this line notes that "glacier margins" were digitised and then quotes data on "total exposed-ice area". This raises a number of questions that need clarifying. Are the digitised glacier margins (presumably the entire glacier) the same as the total exposed ice area? If so, it implies that there is no debris cover on these glaciers? Is this correct? The authors should be very clear what they have actually mapped, both here and in the Methods. This issue permeates much of the manuscript.

INTRODUCTION

P1, line 23: The term "non-climatic forcing" is somewhat confusing. Glacier mass balance can only be 'forced' by climate. Non-climatic factors might then modulate the response of the glacier to a forcing, but it's unclear what the authors mean by non-climatic "forcing". Please re-word or clarify.

P1, line 25: "area of exposed ice" is mentioned here again, but it is unclear whether this is the entire glacier area (i.e. including debris-cover) or not? Further detail is required.

STUDY AREA:

In section 2.1, all of these place names mentioned should be labelled on Figure 1, which is lacking key information (see comments below under Figure 1).

P2, line 8: consider deleting "Sandwiched"

Section 2.3 should include a brief review of previous work on glacier change in the region. As it stands, it implies that virtually nothing has been done in this region, but several papers are cited and discussed later in the manuscript that should first be discussed here. This is important because it will help the reader see how the present studies builds on previous work. It will also give clear credit to previous work.

METHODS

In section 3.1 (data sources), there is no mention of any earlier imagery, e.g. from the 1980s, or 1990s. I suspect most readers will wonder why no attempt was made to collect data from earlier dates to increase the temporal resolution of the data. Thus, it would be very helpful to state why only 2 time-steps were chosen and why these specific time-steps were chosen. i.e. 2000 and 2014.

Section 3.1 should also provide some comment on the vertical and horizontal resolution of the SRTM data and its accuracy in this region.

P3, line 31: the authors note that semi-automated methods "often underestimated the exposed ice extent". How do they know that? Compared to what objective measure? Are there any detailed maps of individual glaciers that could be used as a more objective measure? Further detail is required.

P4, line 14: change "an actual glacier" to "a glacier"

P4, line 20: I think it is "DeBeer" not "Debeer"

P4, line 28: "exposed ice". See above. It is still not clear whether this means the total glacier area or not. Moreover, lines 31-32 make it clear that some glaciers do have debris-cover. Throughout the manuscript, this needs clarifying, i.e. what have they actually mapped?

P5, line 1, I'm surprised that it is possible to use satellite imagery to define debris cover to the nearest cm. Is this correct? With what (un)certainty?

P5, line 9: what elevation does the reanalysis data relate to? Isn't it the case that you can obtain data from different heights above the surface?

RESULTS P5, line 26: It's noteworthy that so many glaciers have disappeared in just 14 years. It would be really nice to show an example of this disappearance in a Figure. Because the authors fail to show some detailed mapping, the methods and results are somewhat lacking in transparency. Indeed, the authors might also want to consider making their dataset more widely available (this is a requirement in some journals).

Section 4.2 – I wonder if it might be better to show how the distribution of glacier areas have changed through time, i.e. show a histogram of binned areas in 2000 to 2014?

P6, line 23: "does not", rather than "doesn't"

DISCUSSION

P7 – many of the papers cited here relating to previous work should be properly introduced in the section on the Study Area (see comments above)

P7, lines 6-10: I find it surprising that two inventories can be so different. This is somewhat brushed under the carpet as a function of the semi-automated techniques misclassifying snow patches as glacier ice. This may be the case, and it would be really nice to show a Figure that illustrates this potential misclassification (or at least shows us some close-ups of how the two outlines compare for different glaciers), but is debris cover also an issue? This manuscript only presents data on "exposed ice", I think. Is this a part of the explanation? Some further discussion and illustration would really help the reader.

P7, lines 19: the recent increase in precipitation is, indeed, notable. I was surprised that there was no discussion of the form of the precipitation, i.e. increases s in ppt in spring, summer and autumn are likely to be dominated by rain (?), which may not impacting on glacier mass balance to the same degree as winter snow. The discussion here seems to be rather brief too simplistic and could do with some more detailed

explanation.

P8, line 8 – this sentence needs re-writing

P8, line 11 – this sentence also needs re-writing. Suggest you split the sentence at "however, "

P8, line 24 – again, debris cover is mentioned, leading to some confusion as to what the authors have actually mapped in this study

P8, line 28: is there any indication as to when this volcanic activity might have taken place?

P8, line 30: again, this mention of surge-type glaciers should have been included in the Study Area chapter. At this point in the manuscript, some further detail on evidence for/against surge-type glaciers would be helpful. Even if you do not have the data that captures a surge, there might be other evidence on the imagery (see Copland et al., 2003: Annals of Glaciology)

CONCLUSIONS:

P9, line 10: It's not possible to detect a "notable acceleration" from the two time-steps. I think the preceding sentence needs to make it clear over what time-steps the rates of change in brackets apply to.

P9, line 30: this is an interesting point made earlier in the manuscript, but there needs to be some appreciation of the altitude of these small glaciers. If most of them are at high elevations, they may not necessarily disappear. Perhaps explain how many of the small glaciers are at lower elevations and are most vulnerable.

FIGURES AND TABLES:

Table 1: it would be very useful if this Table also included the scene IDs for each image.

Table 2: as noted above, this table could easily be supplemented with some histograms

of glacier area distribution at each time-step.

Figure 1: this figure is lacking some important information. Please add key place names that are mentioned in the text and locate Kamchatka on the inset map. I'm also puzzled why the authors do not used a coloured hill-shaded rendition of the DEM.

Figure 2: at this scale, it is almost impossible to see why the different methods are in disagreement. It would be much better to show a close up, maybe as a panel 'b' to this Figure? This will allow the authors to more clearly illustrate what they are mapping and why other methods misclassify the glaciers

Figure 3a: the data on glacier change are far too small on this Figure. I suggest it fills a whole page and maybe even includes some nice close-ups from areas of interest

Figure 3b: this histogram is useful, but the extreme growth of one glacier is interesting. Did this really occur? Can it be illustrated on a Figure and can it be explained? It seems odd to ignore any explanation.

Figure 4: for context, can the outline or location of this figure be shown on one of the earlier Figures, e.g. Fig. 3a or 1?

Figure 5 – the caption on panel (a) says "annual change" – what does this mean? Is this the annual data? Assembled from what? Daily means, monthly means?

Figure 6b and d – what does glacier relief mean? Elevation? Elevation of what (min, mean, max)?

---

## Referee Comment (RC2) · Anonymous Referee #2 · 16 May 2016

STRENGTHS:

Positively 1 - A huge amount of work behind the paper

Positively 2 - A good review of associated publications, including Russian authors

WEAKNESSES:

- The paper is too long. The text contains lots of unnecessary naturally clear details and numbers which hinder its perception. Most numbers should be removed and put in the tables.

- The authors don't seem to take into account the uniqueness of Kamchatka region as an area where Holocene volcanism exists along with glaciation and can not help affecting it. The accomplished analysis is of a "standard" (formal) type, the same as for

non-volcanic regions.

However, along with tephra of recent eruptions (mentioned in the paper) many glaciers in Kamchatka (or their tongues) which motion is permanently destroying the friable slopes of Holocene volcanoes appear to be buried under a thick layer of deposits and continue flowing with it.

At the same time they are nothing to do with rock glaciers having usually a snow-covered accumulation zone and a high flow velocity. In other words, they keep all glacial features but have a thick cover of deposits on themselves which exceeds obviously 5 cm, the threshold used in the paper for digitizing of glacial margins, which seems to be totally invalid for glacial mapping in Kamchatka. As a result the margins of most glaciers where the described effect is observed were digitized incorrectly and should be reconsidered by the authors.

This conclusion is substantially based on the provided image of Tolbachik Volcano (Figure 2) where none of the glacial margins (neither conducted with semi-automated nor digitized manually) show the real extent of glaciers in the area (accepted and used by other authors, including USSR Glacier Inventory which data is used for comparison in the paper).

This is a serious drawback which apparently can affect the results of the accomplished statistical analysis and the paper's conclusions in general.

Now the conclusion that volcanic activity has no evident effect on glacier fluctuations seems to be very questionable and should be reconsidered using a new dataset with redigitized glacial margins.

- Some glaciers appeared to increase their size. No possible explanations are presented.

SMALLER REMARKS:

Kamchatka Peninsula is of particular interest because investigation of its recent glacial

history has been limited (c.f., Khromova et al., 2014; Earl and Gardner, 2016)

No overview or references to the glacial studies which were carried out.

Mass-balance and paleoglaciological campaigns in Koryto glacier (Kronotskiy Peninsula); ice core boring of the crater glacier (Ushkovsky volcano); remote-sensing studies etc.

The peninsula has been shaped by both volcanic and glacial forces (Braitseva et al., 1995, 1997; Ponomareva et al., 2007, 2013; Barr and Clark, 2012 a,b)

There are also folded mountains in Kamchatka by the way.

Subsequent estimates of the total number and surface area of glaciers on the peninsula tend to differ. For example, Muravyev (1999) reports 448 glaciers, with a total area of $\sim 906$ km2, while Solomina et al. (2007) report 446 glaciers, with a total area of $\sim 900$ km2.

But you have to know that for a half of the glaciation area peninsula (Sredinny Range) data were provided from (Vinogradov, 1968) as new qualitative cartographic and satellite materials appeared on this area only in 21 centuries. At the same time for regions of an active volcanism the results leaning on the last decades of researches and for them for passed half a century any are given glacial retreating (especially "Rapid . . .") wasn't observed. Moreover, some glaciers in Klyuchevsky and Avacha groups of volcanoes continued the approach since last century, and, just, at the expense of the languages blocked by a moraine unaccounted by authors.

According to the latest published data on fluctuations of glaciers on the Kamchatka peninsula with use of aero-photo, good permission of space images, cartographic materials, and using of GPS receivers in field researches, really, some parts of modern glaciation retreating are observed. The truth this situation is peculiar mainly for glaciers of nonvolcanic areas. Glaciers on volcanoes generally are in a steady state, or tend to growth.

For example:

– about glaciers of Koryaksky volcano – volcanic region (Manevich, Samoilenko, 2012): "Now there are seven glaciers with total area 8,36 km2. Three of them advance, two are in stationary state and one degrades;

– glaciation of Sredinny Range (Muraviev, Nosenko, 2013): ... in the northern part of the Middle (Sredinny) Range from 1950s to 2002 – diminished by 16.6%;

– glaciation of the not-volcanic regions (Muraviev, 2013): Glaciers on the Kronotsky Peninsula (Eastern Kamchatka) and the Alney-Chashakondzha massif (Sredinny Range) shrink. Since 1950 to 2013 area of Kronotsky glaciers reduced by 18.8 km2, or by 22.9% (for glaciers with areas larger 0.5 km2). Area of the Alney-Chashakondzha glaciers reduced for the period since 1950 to 2010 by 11.6 km2, or by 19.2%, that is comparable to similar characteristics of glacier systems of Altai, Tien Shan, and Caucasus, and this contraction correlates with changes of basic climatic variations, i.e. rising of summer air temperature and decreasing of solid precipitation.

– the glaciers of Avachinskaya volcano group (Manevich et al., 2015): "The present-day glaciation of the Avachinskaya group was analyzed using data obtained during field works of 2007-2010. Twenty seven glaciers were founded the volcanic slopes with their total area of 24.04±3.6 km2. ... Comparison of recent data with aerial photographs of 1974 has allowed estimating changes of the glacier front positions and to reconstruct the glacier dynamics for the last 40 years. Eighteen glaciers have been found to be in the stationary state, seven glaciers advance, and two glaciers degrade."

– (Golub, Rassokhina, 2015): 1. Kropotkin's glacier (Eastern Kamchatka) since the end of the 1970th years of the XX century by 2014 (45 years) receded in the central part of the front in a place of its bifurcation on two tongues on 290 m, the left - on 390 m, the right - on 280 m. The area of a glacier was reduced by 34%; speed of retreating of Kropotkin glacier is increased, since 2003, that is caused by growth of summer air temperatures; thus, from 1976 to 2014 the area of glacier decreased from 0,85 sq.km
to 0,29 sq.km, or for 34%, from which ∼2/3 of receding fall on the first 1.5 decades of 21 centuries.

As far as I know, the russian glaciologists prepare publications on new inventory of glaciers of Klyuchevskaya group of volcanoes, etc., where the stable condition of glaciers of volcanic areas of the peninsula and retreating of glaciers non-volcanic areas similar to a velocity of retreating of other mountains glaciation of Eurasia are recorded.

OTHER REMARKS:

Glacier size (area, perimeter, length and relief) . . .

What is relief is totally unclear? There is no reference to the parameter is the part describing which glacial parameters were used in the statistical analysis.

For each glacier identified in the inventory in 2000, maximum, minimum and median altitude and mean surface slope were calculated from the SRTM DEM, and generalised glacier aspect was estimated from a line connecting the glacier's maximum and minimum altitudes. The mean annual receipt of solar radiation at the surface of each glacier was calculated using the Solar Radiation tool in ArcGIS (algorithms developed by Fu and Rich, 2002), and glacier length was estimated along inferred flowlines.

Glaciers range in length from 170 m to 9930 m, while ranging in altitude from 273 m to 4407 m (a.s.l.), with a mean altitudinal range of 419 m, and a mean surface slope of 16.9°. Rounded numbers should be provided.

GENERAL SUGGESTIONS:

1. The paper is to be shortened getting rid of unnecessary details describing the apparent parts of methodology. More stress should be put to the interpretation of the analysis results. Probably different glacial areas of peninsula can also be analyzed against each other.

2. The statistical analysis should be done once again using the redigitized glacial

margins, delineating the glacial parts covered by debris.

3. The conclusions should be tested with the new redigitized dataset.

Considering the inevitable errors in such studies these two sources rather support than contradict other. The setting off is inappropriate here (Fig. 1).

REFERENCES:

Manevich T.M., Samoilenko S.B. Glaciers of the Koryak volcano // Ice and Snow. No.3 (119). 2012. 25-30 (Summary in English).

Muraviev A.Ya. Nosenko G.A. Glaciation change in the northern part of the Middle Range on the Kamchatka Peninsula in the second half of XX century // Ice and Snow. No.2 (122). 2013. 5-11 (Summary in English).

Muravyev A.Ya. Glacier size changes in Kronotsky Peninsula and Alney-Chashakondzha Massif, Kamchatka Peninsula in the second half of XX century and the beginning of XXI century // Ice and Snow. No.2 (126). 2014. 22-28 (Summary in English).

Muraviev A.Ya., Nosenko G.A. Glaciation change in the northern part of the Middle Range on the Kamchatka Peninsula in the second half of the XX century // Ice and Snow. 2013. âĎŰ 2(122). P. 5–11 (Summary in English).

Manevich T.M., Murav'ev Ya.D., Samoilenko S.B. Glaciers of the Avachinskaya volcano group: current condition (state) // Ice and Snow. Vol.55. No.3 (119). 2015. 14-26 (Summary in English).

Golub N.V., Rassokhina L.I. The dynamics of the glaciers at Bolshoi Semyachik volcano (Eastern Kamchatka) at the beginning of the 21st century and formation of vegetation on the young moraines // Bull. Of Kamchatka regional association "Educational-Scientific Center". Earth Sciences. Vol.28, No.4. 2015. 60-71 (Abstract in English).

[Figure]

[Figure]

**Fig. 1.** The margins of the Tolbachik volcano's glaciers proved by field studies in the area

---

## Author Comment (AC1) · 31 May 2016

Our response and associated actions relating to SC1's comments:

SC1.1

Comment: According to Figure 2 of the proposed paper, both semi-automatic methods are not suitable (or incorrectly used) for identification of glaciers in the zones of modern volcanic activity, the major part of which is covered with a ground moraine. This person, who manually interpreted the images, has determined not the glacial boundary, but the boundary of surface of open firn and ice. The scheme 1 (see the attached pdf) shows results from determination of glacial boundaries on Tolbachik Volcano based on space images ASTER 19.07.2012, GeoEye-1 01.09.2011 and GeoEye-1 04.07.2013. Data from field observations was used for determination (GPS points and tracks, photos).

[Figure]

In order to demonstrate how data of interpretation fit the real setting the images below show glaciers (mainly their terminus) and observation sites.

Response: As noted in the original manuscript, our aim was to analyse variations in the extent of 'exposed' glacial ice. This focus on exposed ice was adopted because of difficulties with accurately delimiting debris covered margins, and is an approach used elsewhere in eastern Russia (see Stokes et al., 2013, cited in the manuscript). As a result, the reviewer is correct in stating that for debris covered glaciers (which are largely restricted to volcanically active regions of Kamchatka) our original mapping did not reflect the full glacier boundary.

Action: In light of this (and because of similar points are raised by other reviewers), we have now updated the manuscript to include the data for full glacier surface areas, and now use the term "glacier" rather than "exposed ice". The resulting impact on the data presented in the revised manuscript is detailed in the table (see supplementary figure 1).

As seen from the table (supplementary figure 1), including debris covered portions of glaciers has some impact on the statistics quoted (these values are have been updated throughout the manuscript), but this has little impact on the study conclusions, since the statistical significance of relationships between attributes does not really change (though all relevant tables and figures have been updated). The only exception to this is that the relationship between glacier aspect and glacier area change is now significant (though a weak relationship). As a result some of the text has been altered. For example, the penultimate sentence in section 4.4., which now reads:

"Glaciers are predominantly found with an aspect-bias towards northerly and western directions (54.98 %), and glacier aspect shows a weak ($r = 0.15$), but statistically significant, relationship with changes in glacier surface area, though only when expressed in km2, rather than as a percentage. Similarly, analysis of insolation patterns reveals a weak, but statistically significant, correlation ($r = 0.18$) with change in glacier surface

area, but only when expressed in km2, rather than as a percentage."

Rather than (from the original manuscript): "Glaciers are predominantly found with an aspect-bias towards northerly and western directions (48.96%), but aspect doesn't show a statistically significant relationship with changes in glacier surface area (when expressed as total area, or as percentage change). 25 Analysis of insolation patterns reveals a weak, but statistically significant, correlation (r = 0.19) with change in glacier surface area, but only when expressed in km2, rather than as a percentage."

Section 5.3.3., has also been updated to read: "The aspect bias exhibited by Kamchatkan glaciers, combined with the statistically significant relationship between area loss and total insolation indicates that glaciers exposed to most solar radiation typically show a greater reduction in their overall surface area (see Evans, 2006). However, though statistically significant, the relationship between glacier aspect and changes in glacier surface area is comparatively weak, suggesting that local variations in insolation (e.g., related to topographic shading) are likely important in protecting glaciers from recession (see Paul and Haeberli, 2008; Stokes et al., 2013)"

Rather than (from the original manuscript): "The aspect bias exhibited by Kamchatkan glaciers, combined with the statistically significant relationship between area loss and total insolation indicates that glaciers exposed to most solar radiation typically show a greater reduction in their overall surface area, (see Evans, 2006). However, the lack of a statistically significant relationship between glacier aspect and changes in glacier surface area, suggest that local variations in insolation (e.g., related to topographic shading) are likely important in protecting glaciers from recession (see Paul and Haeberli, 2008). Note, the lack of any clear aspect control on glacier fluctuations has also been observed in other regions globally (Debeer and Sharp, 2007; Tennant et al., 2012), and is often considered to reflect the importance of other factors (e.g., local shading, debris cover, etc.) in modulating glacier recession 25 (Stokes et al., 2013)."

The final sentence of conclusion 3 has also been updated to read: "However, though

statistically significant, the relationship between glacier aspect and changes in glacier surface area is comparatively weak, suggesting that local variations in insolation (e.g., related to topographic shading) are important in regulating fluctuations of Kamchatka's glaciers."

Rather than (from the original manuscript): "However, glacier aspect fails to show a statistically significant relationship with changes in glacier surface area, suggesting that local variations in insolation (e.g., related to topographic shading) are important in regulating fluctuations of Kamchatka's glaciers."

Given the role of debris in regulating glacier behaviour, we now refer to this more explicitly in the manuscript. Specifically, in section 4.4., we add the following paragraph:

"Due to their location on the slopes of volcanic peaks, many glaciers in the Central Kamchatka Depression have tongues covered with significant debris accumulations (rock and ash) (see Yamaguchi, et al., 2007). In total, based on mapping in 2000, $\sim$ 8% (n = 52) of the peninsula's glaciers are classed as debris-covered. When the glacier population is split into 'debris-covered' and 'non debris-covered' samples in this way, the former lose on average $\sim$ 11% (0.62 km2) of their surface area between 2000 and 2014, while the latter lose $\sim$ 44% (0.24 km2). Thus, over the 2000–2014 period, debris-covered glaciers lose less surface area than non debris-covered examples, though this relationship is only statistically significant when values are expressed as a percentage, rather than in km2."

We re-name section 5.3.4., from "Other non-climatic factors" to "Volcanic controls and debris cover". This section is also re-written to emphasise the role of debris cover, and now reads:

"Across the peninsula, 165 glaciers are located within a 50 km radius of an active volcano, with 292 glaciers within a 100 km radius. During the period of observation (i.e., 2000–2014), > 40 individual volcanic eruptions were documented on the peninsula (VONA/KVERT, 2016). Although, there is evidence that some glaciers were covered

by tephra as a result of these eruptions, there is no evidence to suggest that this had a discernible influence of glacier fluctuations during the period of observation. This likely reflects the longer response time of glaciers to tephra deposition, since a number of Kamchatka's glaciers are known to have responded to volcanic ash cover over longer time periods (see Barr and Solomina, 2014). It is also apparent, from the present study, that debris-covered glaciers on the Peninsula are typically less responsive to external forcing, since they lose less surface area (both in absolute and, particularly, relative terms) than non debris-covered examples. This likely reflects the insulating effect of the accumulated surface debris (Vinogradov et al., 1985). In addition to the influence exerted by surface debris, some of Kamchatka's glaciers are known to be of 'surge-type', with surface indicators of past surge activity (e.g., looped moraines and heavy crevassing) (see Copland et al., 2003), and with documented surges during the 20th Century (as noted sect. 2.3) (Vinogradov et al., 1985; Yamaguchi et al., 2007). However, there was no evidence of surging during the period of observation. This might reflect the comparatively short time period considered, or may be a reflection of volcanically controlled surging (see section 2.3) (Muraviev et al., 2012), which therefore lacks periodicity."

Rather than (from the original manuscript):

"Across the peninsula, 165 glaciers are located within a 50 km radius of an active volcano, with 292 glaciers within a 100 km radius. Although, there is evidence that some of these glaciers were covered by tephra as a result of proximal volcanic activity, there is no evidence to suggest that this had a discernible influence of glacier fluctuations during the period of observation. In addition, due to their volcanic setting, some of Kamchatka's glaciers are known to be of 'surge-type' (Vinogradov et al., 1985; Yamaguchi et al., 2007). However, there was no evidence of surge activity during the period of observation, perhaps reflecting the comparatively short time period considered."

Debris cover is now also mentioned in the final sentence of the abstract, which now reads: "Analysis of possible controls indicates that these glacier fluctuations were likely

governed by variations in climate (particularly rising summer temperatures), though the response of individual glaciers was modulated by other (non-climatic) factors, principally glacier size, local shading and debris-cover."

Rather than (from the original manuscript):

"Analysis of possible controls indicates that these glacier fluctuations were likely governed by variations in climate (particularly rising summer temperatures), though the response of individual glaciers was modulated by other (non-climatic) factors, principally glacier size and local shading."

SC1.2

Comment: Fine scale diagram in Figure 1 of the paper does not allow adequate assessment of quality of determined glacial boundary of the volcanic massif with Klyuchevskoy, Kamen, Plosky Blizhny and Plosky Dalny volcanoes as well as Sheveluch Volcano. The authors suppose that the contour shape of the zones occupied by the glaciers in 2000 is enough for conclusion similar to that in paragraph 1 where the authors have determined not the glacial boundary, but the boundary of the surface of open ice and firn. From that follows the conclusion – in this paper, the glaciation area of active Kamchatka volcanism by 2000 is more than twice understated. There is good reason to believe that the situation with the 2014 satellite images interpretation is not better. Ice and firn area size variations on satellite images are caused by many factors, e.g. accumulated seasonal snowfalls, atmospheric temperature during the summer, summer snow flurries etc. It is impossible to determine both the boundary and area of glaciers in general using these area size variations of glacial parts, which are not covered with moraines. The article does not contain the shooting dates for used satellite images. The dates of shooting are important for understanding how the setting of survey corresponds to that in-situ by the end of the ablation period, i.e. the understanding of how the remaining seasonal snow cover and the snow beds could corrupt the results of glacial boundaries determination (especially in the accumulation area).

Probably, the errors caused by the remaining seasonal snow cover and the snow beds are the reason for big difference of glaciers areas (as in Figure 4) and their amount between the 2000 and the 2014 images measurements.

Response: The date of acquisition ('shooting dates') for each of the satellite images used in this investigation is presented in Table 1 of the manuscript (in both the original and revised manuscript). As can be seen from Table. 1, each of our images was captured during the ablation season, thus limiting the potential for erroneously mapping snow as exposed glacial ice. The reviewer is correct to emphasise that snow cover can lead to erroneous estimates of glacial retreat. This is something we discuss (and address) in sections 3.2 and 3.3 of the manuscript ((in both the original and revised versions). The reviewer is correct to state that only mapping 'open ice' results in an underestimation of glacier area (particularly in areas of active volcanism).

Action: In light of the reviewer's concern, Table 1 has now been updated to include scene IDs to provide additionally background information on the satellite imagery used. In order to provide more information about glacier boundaries on volcanic massifs, figure 3 (C) and (D) (in the revised manuscript) now show mapped margins of glaciers in the Klyuchevskaya Volcanic Group and on the slopes of Ichinsky Volcano. We have addressed the reviewer's concerns about 'open ice' vs. debris covered ice in our response and actions to point SC1.1 (i.e., our data are now based on total glacier area, rather than exposed ice alone).

SC1. 3

Comment: In comparison with the data of other authors the amount of glaciers found in Kamchatka by authors is underestimated. The authors incorrectly determine the boundaries of the glaciers due to insufficient knowledge of the investigated glaciation area specific features, the lack of the accuracy of the used satellite images and the use of determination methods not valid for glaciers with high debris cover (or misuse of quite valid methods). During the investigation the authors reveal significant reduction

of the glaciers area and the increase in their amount from 2000 to 2014, probably explaining by measurement errors resulted from the following factors: the presence of the remaining seasonal snow cover and snow patches at the time of shooting; the lack of the accuracy of the used satellite images Landsat and ASTER for small (<0.5 kmЁЕ2) glaciers; incorrect methods of determinations used by the authors. This study does not reflect the changes in glaciation of Kamchatka from 2000 to 2014. The results are based on invalid data.

Response: In response to the suggestion that we 'incorrectly determine the boundaries of glaciers' we reiterate that in the original manuscript our focus was on 'exposed ice' (mentioned 9 times in the original manuscript). However, we accept that this value underestimates the total glacier area. As noted in SC1.1., we now map total glacier area (including debris). The results of relevance here are that, by including debris covered areas: (i) our estimates of total glacier area increase (for both 2000 and 2014); (2) the estimate of area loss between 2000 and 2014 is reduced, but remains considerable; (3) the increase in the total number of glaciers from 2000 to 2014 is reduced slightly (from 90 in the original manuscript, to 65 in the revised version). As alluded to be the reviewer, the resolution of the satellite imagery does create some difficulty with mapping small ice masses, however, Landsat and ASTER imagery are the only data to allow repeated multi-year analysis of these glacier margins, and have been widely used to map glaciers of similar dimensions in other regions globally (e.g., Stokes et al., 2013). Additionally (as noted in the manuscript), we applied a threshold of 0.02 km2 (see section 3.2) which has been used throughout the analysis. This is above the threshold of 0.01km2 defined by Paul et al., (2009) (cited in the manuscript) as the lower limit for discerning glacier ice using Landsat imagery.

Action: This is largely dealt with under SC1.1. Issues relating to the timing of image capture (and difficulties with snow cover) are addresses under SC1.2.

Our response and associated actions relating to RC1's comments:

[Figure]

RC1.1

Comment: This paper presents data on areal changes of comprehensive dataset of glaciers on the Kamchatka Peninsula, obtained using satellite imagery from 2000 and 2014. It reveals a widespread reduction in glacier area, which is compared to older inventory data and to a more recent inventory. The paper then explores possible correlations between glacier shrinkage and climate reanalysis data, in addition to glacier characteristics (e.g. size, elevation, aspect, etc.). The paper is concise (perhaps too concise), well-written and well-organised, with some helpful Figures and Tables (albeit that could be improved and supplemented). It presents an impressive dataset and, given that this region is comparatively poorly studied, I'm generally very supportive of publication in The Cryosphere. However, I would suggest that there are a number of issues that would need to be resolved and clarified prior to publication, which I detail below.

The most important and serious of these issues is that the authors need to be much clearer about what they have actually mapped and how this compares to other approaches and previous inventories. The authors appear to have mapped "exposed ice", but then make very little comment on how much debris might be obscuring glaciers in this region. In addition to some additional clarification and discussion, it is important that the authors provide some detailed comparisons of their mapping with other approaches. As it stands, the methods and results are somewhat lacking in transparency.

Response: We acknowledge (and are grateful for) these useful and valid comments.

Action: The issues raised here are largely dealt with in the sections which follow. The reviewer's concerns about 'exposed' vs. debris-covered ice are addressed in our response to SC1.1. In relation to the suggestion that the paper might be 'too concise', please see our response to RC2.1.

RC1.2

Comment: P1, line 9: consider deleting "With this in mind"

Action: As suggested, this has been deleted.

RC1.3

Comment: P1, line 10-12: this line notes that "glacier margins" were digitised and then quotes data on "total exposed-ice area". This raises a number of questions that need clarifying. Are the digitised glacier margins (presumably the entire glacier) the same as the total exposed ice area? If so, it implies that there is no debris cover on these glaciers? Is this correct? The authors should be very clear what they have actually mapped, both here and in the Methods. This issue permeates much of the manuscript.

Action: We have now addressed this by expanding the analysis to include the debris covered areas, and the term 'exposed ice' is removed from the manuscript (since the whole glacier is now mapped). This is outlined in detail in our response to SC1.1.

RC1.4

Comment: P1, line 23: The term "non-climatic forcing" is somewhat confusing. Glacier mass balance can only be 'forced' by climate. Non-climatic factors might then modulate the response of the glacier to a forcing, but it's unclear what the authors mean by nonclimatic "forcing". Please re-word or clarify.

Action: The terms "non-climatic forcing" and "non-climatic control" have now been replaced with "non climatic modulation" throughout the manuscript.

RC1.5

Comment: P1, line 25: "area of exposed ice" is mentioned here again, but it is unclear whether this is the entire glacier area (i.e. including debris-cover) or not? Further detail is required.

Action: This is addressed in our response to SC1.1.

RC1.6

Comment: In section 2.1, all of these place names mentioned should be labelled on Figure 1, which is lacking key information (see comments below under Figure 1)

Action: The place names mentioned in the text are now labelled in figure 1.

RC1.7

Comment: P2, line 8: consider deleting "Sandwiched"

Action: As suggested, this has been deleted.

RC1.8

Comment: Section 2.3 should include a brief review of previous work on glacier change in the region. As it stands, it implies that virtually nothing has been done in this region, but several papers are cited and discussed later in the manuscript that should first be discussed here. This is important because it will help the reader see how the present studies builds on previous work. It will also give clear credit to previous work.

Action: This has now been addressed by adding the following paragraph to section 2.3: "In addition to these peninsula-wide inventories, recent glacier fluctuations in specific regions of Kamchatka have been conducted (locations identified in Fig. 1). For example, based on the analysis of satellite imagery, data from the UGI, and aerial photographs, Muraviev and Nosenko (2013) identified a 16.6% decline in glacier surface area in the northern sector Sredinny Range between 1950 and 2013. Muraviev (2014), using these same techniques, identified a 19.2% decrease in glacier surface area in the Alney-Chashakondzha region between 1950 and 2013 and a 22.9% decrease on the Kronotsky Peninsula between 1950 and 2010. On the basis of field-work (2007–2010) and the analysis of aerial photographs (from 1974), Manevich et al. (2015) documented fluctuations of 27 glaciers on the slopes of the Avachin-skaya Volcanic Group between 1974 and 2010, and found that, during this period, seven glaciers advanced, two retreated, and eight remained largely stationary. Other,

smaller scale studies include investigations of glaciers on the slopes of Ichinsky Volcano (55.690°N, 157.726°E) (see Matoba et al., 2007), Koryak Volcano (53.321°N, 158.706°E) (Manevich and Samoilenko, 2012), in the Klyuchevskaya Volcanic group (∼ 56.069°N, 160.467°E) (see Shiraiwa et al, 2001), and at Koryto glacier (54.846°N, 161.758°E) on the Kronotsky Peninsula (see Yamaguchi et al., 1998, 2007, 2008). As a result of these investigations, some of the glaciers in the Central Kamchatka Depression have been documented as 'surge-type' (Vinogradov et al., 1985). For example, Bilchenok glacier (56.100°N, 160.482°E) in the Klyuchevskaya volcanic group, is known to have surged ∼ 2 km in 1959/1960 and again in 1982/84 (Muraviev et al., 2012). In this example, surging appears unrelated to climate, and is likely driven by the strengthening of seismic activity at Ushkovsky Volcano (56.069°N, 160.467°E), upon which the glacier sits (Muraviev et al., 2012). Direct mass balance observations from Kamchatka's glaciers are limited, with the longest continuous record, obtained for Kozelsky glacier (53.245°N, 158.846°E), spanning the 1973–1997 period. Other mass balance measurements, from Koryto, Mutnovsky SW (52.448°N, 158.181°E), Mutnovsky NE (52.460°N, 158.220°E), and Kropotkina (54.321°N, 160.034°E) glaciers, are far shorter and often discontinuous (see Barr and Solomina, 2014)."

RC1.9

Comment: In section 3.1 (data sources), there is no mention of any earlier imagery, e.g. from the 1980s, or 1990s. I suspect most readers will wonder why no attempt was made to collect data from earlier dates to increase the temporal resolution of the data. Thus, it would be very helpful to state why only 2 time-steps were chosen and why these specific time-steps were chosen. i.e. 2000 and 2014.

Response: This is a valid point and older imagery was analysed, however full coverage of the Peninsula's glaciers (with minimal snow and cloud cover) was only possible for 2000 and 2014.

Action: To clarify this issue, we have added the following to section3.1: "Satellite images captured prior to 2000 (e.g., Landsat MSS and CORONA) were obtained and analysed, however images captured during the ablation season, with minimal snow- and cloud-cover were limited, and full coverage of the peninsula's glaciers was lacking. As a result, the present study only focuses on 2000 and 2014 (i.e., those years with full satellite coverage of the region's glaciers)."

RC1.10

Comment: Section 3.1 should also provide some comment on the vertical and horizontal resolution of the SRTM data and its accuracy in this region.

Action: To address this, the final sentence of section 3.1., now reads: "To analyse glacier topography, elevation data was obtained from the SRTM Global digital elevation model (DEM). This reflects the surface topography of the glaciers in 2000, with a horizontal resolution of $\sim$ 30 m, and with vertical errors < 16m (at the 90% confidence level) (Farr et al., 2007)."

Rather than (from the original manuscript): "To analyse glacier topography, elevation data was obtained from the SRTM Global 30m digital elevation model (DEM) (reflecting the surface topography of the glaciers in 2000)."

RC1.11

Comment: P3, line 31: the authors note that semi-automated methods "often underestimated the exposed ice extent". How do they know that? Compared to what objective measure? Are there any detailed maps of individual glaciers that could be used as a more objective measure? Further detail is required.

Action: To address this issue, we have removed "often underestimating the exposed ice extent" from this sentence, which now reads: "The semi-automated techniques (specifically a RED/SWIR ratio and Normalised Difference Snow Index (NDSI)) produced rapid outputs and were easily applied across large regions, but had difficulty differentiating ice from the surrounding environment (Fig. 2)."

Notably, we now point the reader to Fig. 2 here. Fig. 2 in the revised manuscript replaces the original. In the revised version, differences between the mapping methods are easier to differentiate (since we focus on a smaller area). The revised figure also points the reader to specific locations where semi-automated techniques erroneously map snow patches and fail to map shaded sections of glacial ice. To highlight this, the figure caption now reads:

"Figure 2: Comparison of glacier margins (on the slopes of Mutnovsky Volcano) delineated by two semi-automated techniques and manual mapping (Landsat 2014 background image). This figure illustrates how manual mapping can help identify snow patches and shaded areas of glacier ice, thereby allowing the former to be discounted from, and the latter included in, the glacier inventory."

Rather than (from the original manuscript): "Figure 2: Comparison of glacier margins delineated by two semi-automated techniques and manual mapping".

RC1.12

Comment: P4, line 14: change "an actual glacier" to "a glacier"

Action: As suggested, this has now been changed.

RC1.13

Comment: P4, line 20: I think it is "DeBeer" not "Debeer"

Action: As suggested, this change has now been made in the text (in 3 separate places) and reference list.

RC1.14

Comment: P4, line 28: "exposed ice". See above. It is still not clear whether this means the total glacier area or not. Moreover, lines 31-32 make it clear that some glaciers do have debris-cover. Throughout the manuscript, this needs clarifying, i.e. what have they actually mapped?

Action: This is addressed in our response to SC1.1.

RC1.15

Comment: P5, line 1, I'm surprised that it is possible to use satellite imagery to define debris cover to the nearest cm. Is this correct? With what (un)certainty?

Action: Since all debris covered areas are now included in the analysis (see our response to SC1.1.), the sentence in question is no longer applicable to the methods, and has been deleted from the manuscript (as has the associated reference to Ranzi et al., 2004, who are not cited elsewhere in the manuscript).

RC1.16

Comment: P5, line 9: what elevation does the reanalysis data relate to? Isn't it the case that you can obtain data from different heights above the surface?

Action: To clarify this issue, the following sentence has now been added to section 3.4: "The precipitation data reflect surface conditions, and temperature data reflect conditions at a height of 2 m (a.s.l)."

RC1.17

Comment: P5, line 26: It's noteworthy that so many glaciers have disappeared in just 14 years. It would be really nice to show an example of this disappearance in a Figure. Because the authors fail to show some detailed mapping, the methods and results are somewhat lacking in transparency. Indeed, the authors might also want to consider making their dataset more widely available (this is a requirement in some journals).

Action: A new figure showing a glacier which disappears over the period of observation has been added to the revised manuscript (Fig. 3). In relation to 'disappearing' glaciers, we have now edited part of section 4.2., to read: "The increase in glacier number occurred despite the loss of 46 glaciers during this period (these glaciers were all < 0.5 km2 in 2000 and either completely disappeared or fell below the size threshold of 0.02
km2 by 2014) (see Fig. 3), and primarily reflects the fragmentation of larger glaciers"

Rather than (from the original manuscript):

"The increase in glacier number occurred despite the complete disappearance of 42 glaciers during this period (these glaciers were all < 0.5 km2 in 2000), and primarily reflects the fragmentation of larger glaciers".

We have also edited the associated section of conclusion 1, which formerly used the term "complete disappearance" but now refers to "disappearance".

In altering the above sentences, we not only point the readers to figure 3, but also emphasise that a 'disappearing' glacier might fall below the size threshold by 2014, rather than completely disappearing.

The updated version of figure 2 (referred to in our response to RC1.11), and the 'new' figure 3 (now fig 4) panels (C) and (D) (referred to in our response to SC1.2) provide more detailed illustrations of our mapping than in the original manuscript. We feel that these updated figures address the reviewer's concerns about a lack of transparency.

RC1.18

Comment: Section 4.2 – I wonder if it might be better to show how the distribution of glacier areas have changed through time, i.e. show a histogram of binned areas in 2000 to 2014?

Action: We have plotted the data as suggested by the reviewer (see supplementary figure 2). However, we don't feel that the figure is particularly enlightening, and, as a result, it is not included in the revised manuscript.

RC1.19

Comment: P6, line 23: "does not", rather than "doesn't"

Action: The section in question has now been re-worded to address the fact that there

relationship between glacier area change and aspect is weak but statistically significant addressed in response to SC1.1), hence the reviewer's comment (though valid for the original manuscript) is no longer relevant.

RC1.20

Comment: P7 – many of the papers cited here relating to previous work should be properly introduced in the section on the Study Area (see comments above).

Action: As the reviewer suggests, these cited papers (and others) are now introduced in section 2.3. This is explained in detail in our response to RC1.8.

RC1.21

Comment: P7, lines 6-10: I find it surprising that two inventories can be so different. This is somewhat brushed under the carpet as a function of the semi-automated techniques misclassifying snow patches as glacier ice. This may be the case, and it would be really nice to show a Figure that illustrates this potential misclassification (or at least shows us some close-ups of how the two outlines compare for different glaciers), but is debris cover also an issue? This manuscript only presents data on "exposed ice", I think. Is this a part of the explanation? Some further discussion and illustration would really help the reader.

Response: After completing the analysis by including debris covered sections of ice (rather than 'exposed ice' alone) (see details in our response to SC1.1.) the total surface area difference between glaciers in our 2014 inventory and the estimate generated by Earl and Gardner (2016) is now reduced (formerly: 305 km2; now 177 km2) but remains considerable. As noted in the original, in section 5.1., we largely attribute this difference to "their [Earl and Gardner's] semi-automated approach to mapping (i.e., NDSI), which can lead to an overestimation of glacier area as a result of snow patches being erroneously classified as glaciers (Man et al., 2014), combined with the fact that their inventory was generated from a composite of satellite images, meaning their

mapping does not reflect glacier extent during a single specified year (as noted in Sect. 2.3.)."

Action: We include the area for 'debris-covered' as well as 'exposed' ice, which narrows the gap between our surface area estimates and Earl and Gardner's (2016) estimate. We also keep the paragraph in section 5.2., attempting to explain this difference

RC1.22

Comment: P7, lines 19: the recent increase in precipitation is, indeed, notable. I was surprised that there was no discussion of the form of the precipitation, i.e. increases s in ppt in spring, summer and autumn are likely to be dominated by rain (?), which may not impacting on glacier mass balance to the same degree as winter snow. The discussion here seems to be rather brief too simplistic and could do with some more detailed explanation

Action: To address this issue, the sentence in question now reads:

"This recent increase in precipitation is particularly notable during autumn (Fig. 6A), though more importantly for glacier mass balance, winter precipitation also increases during this period, likely contributing directly to increased snow accumulation. The temperature data (Fig. 6B) appears to show a warming trend from the 1950s to the late 1990s, which has continued (with some fluctuations) to the present day. Most significantly, there has been a sharp increase in average summer temperatures (June, July and August) since the early 21st century (see Fig. 6B)."

Rather than (from the original manuscript): "This recent increase in precipitation is particularly notable during autumn (Fig. 5A). The temperature data (Fig. 5B) appears to show a warming trend from the 1950s to the late 1990s, which has continued (with some fluctuations) to the present day. Most significantly, there has been a sharp increase in average summer temperatures (June, July and August) since the early 21st century (see Fig. 5B)."

RC1.23

Comment: P8, line 8 – this sentence needs re-writing

Action: As suggested, this sentence has been re-written, and now reads: "This rapid decline in the area of smaller glaciers on the Kamchatka Peninsula could result in the loss of many over coming decades, as $\sim$ 75% of the glaciers mapped in 2014 have an area < 0.5 km2, of which $\sim$ 87 % have a maximum altitude < 2000 m (a.s.l.), likely making them particularly sensitive to future warming. This supports the view of Ananicheva et al. (2010), who suggest that by 2100, only the largest glaciers on the highest volcanic peaks will remain."

Rather than (from the original manuscript): "This rapid decline in the area of smaller glaciers on the Kamchatka Peninsula could result in the loss of many over coming decades, as $\sim$ 78% of the glaciers mapped in 2014 have an area < 0.5 km2 . This supports the view of Ananicheva et al. (2010), who suggest that by 2100, only the largest glaciers on the highest volcanic peaks will remain."

RC1.24

Comment: P8, line 11 – this sentence also needs re-writing. Suggest you split the sentence at "however,"

Action: As suggested, this sentence has been edited (splitting the sentence at 'however'), and now reads:

"Comparisons between glacier median and maximum altitude and surface area fluctuations would appear to suggest that these factors exert some control on area change. However, the lack of any statistically significant relationship between area loss and minimum altitude might indicate that, rather than exerting a direct control on glacier area, glaciers with high maximum and median altitudes are typically the largest on the peninsula (i.e., there are positive and statistically significant relationships between glacier area and both maximum (r = 0.37) and median (r = 0.25) altitude), and that size

exerts the primary control on glacier behaviour in this relationship."

Rather than (from the original manuscript): "Comparisons between glacier median and maximum altitude and surface area fluctuations would appear to suggest that these factors exert some control on area change, however, the lack if any statistically significant relationship between area loss and minimum altitude might indicate that, rather than exerting a direct control on glacier area, glaciers with high maximum and median altitudes are typically the largest on the peninsula (i.e., there are positive and statistically significant relationships between glacier area and both maximum (r = 0.38) and median (r = 0.29) altitude), and that size exerts the primary control on glacier behaviour in this relationship."

RC1.25

Comment: P8, line 24 – again, debris cover is mentioned, leading to some confusion as to what the authors have actually mapped in this study

Action: This is addressed in our response to SC1.1.

RC1.26

Comment: P8, line 28: is there any indication as to when this volcanic activity might have taken place?

Response: Volcanic activity on the peninsula is almost continuous.

Action: To address this issue, we have added to following sentence to section 5.3.4: "During the period of observation (i.e., 2000–2014), > 40 individual volcanic eruptions were documented on the peninsula (VONA/KVERT, 2016)."

RC1.27

Comment: P8, line 30: again, this mention of surge-type glaciers should have been included in the Study Area chapter. At this point in the manuscript, some further detail on evidence for/against surge-type glaciers would be helpful. Even if you do not have

the data that captures a surge, there might be other evidence on the imagery (see Copland et al., 2003: Annals of Glaciology)

Action: To address this, we have added the following as part of section 2.3 (part of the new section noted in our response to RC1.8): "As a result of these investigations, some of the glaciers in the Central Kamchatka Depression have been documented as 'surge-type' (Vinogradov et al., 1985). For example, Bilchenok glacier (56.100°N, 160.482°E) in the Klyuchevskaya volcanic group, is known to have surged $\sim$ 2 km in 1959/1960 and again in 1982/84 (Muraviev et al., 2012). In this example, surging appears unrelated to climate, and is likely driven by the strengthening of seismic activity at Ushkovsky Volcano (56.069°N, 160.467°E), upon which the glacier sits (Muraviev et al., 2012)."

We have also edited the final sentence of section 5.3.4, which now reads part of the new section noted in our response to SC1.1):

"In addition to the influence exerted by surface debris, some of Kamchatka's glaciers are known to be of 'surge-type', with surface indicators of past surge activity (e.g., looped moraines and heavy crevassing) (see Copland et al., 2003), and with documented surges during the 20th Century (as noted sect. 2.3) (Vinogradov et al., 1985; Yamaguchi et al., 2007). However, there was no evidence of surging during the period of observation. This might reflect the comparatively short time period considered, or may be a reflection of volcanically controlled surging (see section 2.3) (Muraviev et al., 2012), which therefore lacks periodicity."

Rather than (from the original manuscript): "In addition, due to their volcanic setting, some of Kamchatka's glaciers are known to be of 'surge-type' (Vinogradov et al., 1985; Yamaguchi et al., 2007). However, there was no evidence of surge activity during the period of observation, perhaps reflecting the comparatively short time period considered."

RC1.28

Comment: P9, line 10: It's not possible to detect a "notable acceleration" from the two time-steps. I think the preceding sentence needs to make it clear over what time-steps the rates of change in brackets apply to.

Action: To address this, the word 'notable' has been removed and the time steps have been added in to this sentence, which now reads: "By 2014, the total number of glaciers had increased to 738 but their surface area had reduced to 592.9 $\pm$ 20.4 km2 , this suggests an acceleration in the rate of area loss since 2000 (from $\sim$ 0.24–0.29 % a-1 between the 1950s and 2000, to $\sim$ 1.76% a-1, between 2000 and 2014)."

Rather than (from the original manuscript): "By 2014, the total number of glaciers had increased to 766 but their surface area had reduced to 465.1 $\pm$ 15.7 km2 , this suggests a notable acceleration in the rate of area loss (from $\sim$ 0.55–0.67 % a-1 to $\sim$ 2.52% a-1) since 2000."

RC1.29

Comment: P9, line 30: this is an interesting point made earlier in the manuscript, but there needs to be some appreciation of the altitude of these small glaciers. If most of them are at high elevations, they may not necessarily disappear. Perhaps explain how many of the small glaciers are at lower elevations and are most vulnerable

Action: As noted in our response to RC1.23, this has been addressed in section5.3.1 of the revised manuscript, and is now also addressed in conclusion 4, by adding the following: "of which $\sim$ 87 % have a maximum altitude < 2000 m (a.s.l.), likely making them particularly sensitive to future warming."

RC1.30

Comment: Table 1: it would be very useful if this Table also included the scene IDs for each image

Action: As suggested, scene IDs have now been added to Table 1.

RC1.31

Comment: Table 2: as noted above, this table could easily be supplemented with some histograms of glacier area distribution at each time-step.

Action: This is addressed in our response to RC1.18.

RC1.32

Comment: Figure 1: this figure is lacking some important information. Please add key place names that are mentioned in the text and locate Kamchatka on the inset map. I'm also puzzled why the authors do not used a coloured hill-shaded rendition of the DEM.

Response: A grey-scale DEM was chosen as we felt a coloured map took away from the ice extent, which we wanted to highlight in this figure.

Action: As suggested, key place names (mentioned in the text) have now been added to figure 1. In addition, Kamchatka is now located (using a rectangle) on the inset map. We have not changed the colour of the DEM.

RC1.33

Comment: Figure 2: at this scale, it is almost impossible to see why the different methods are in disagreement. It would be much better to show a close up, maybe as a panel 'b' to this Figure? This will allow the authors to more clearly illustrate what they are mapping and why other methods misclassify the glaciers

Action: As noted in our response to RC1.11, to address this issue, in the revised manuscript figure 2 has been replaced with a more detailed example.

RC1.34

Comment: Figure 3a: the data on glacier change are far too small on this Figure. I suggest it fills a whole page and maybe even includes some nice close-ups from areas

of interest

Action: To address this, the data in Fig 4A (formerly fig 3A) are now larger. Also, as the reviewer suggests, we have also include panels (C) and (D) showing 'close-ups' of areas of interest (as noted in our response to RC1.17).

RC1.35

Comment: Figure 3b: this histogram is useful, but the extreme growth of one glacier is interesting. Did this really occur? Can it be illustrated on a Figure and can it be explained? It seems odd to ignore any explanation

Action: To address this issue we have edited the sentence in section 4.2., which direct readers to this image (now fig 4). This text in question now reads: "Of the 17 glaciers that increased in area during this period, the majority experienced minor growth. One glacier increased by $\sim$ 140% (see Fig. 4B), but was very close to the size threshold of 0.02 km2 in 2000, and the area gain over the period of observation only equates to 0.03 km2."

Rather than (from the original manuscript): "Of the 17 glaciers that increased in area during this period, the majority experienced minor growth, though one glacier increased by $\sim$ 140% (see Fig. 3B)"

RC1.36

Comment: Figure 4: for context, can the outline or location of this figure be shown on one of the earlier Figures, e.g. Fig. 3a or 1?

Action: To address this issue, a contextual location map has now been added this figure (fig 5). This is also true of figures 2 and 3.

RC1.37

Comment: Figure 5 – the caption on panel (a) says "annual change" – what does this mean? Is this the annual data? Assembled from what? Daily means, monthly means?

Action: The caption in question (for what is now fig.6.) has now been changed to "Annual Data", rather than "annual change". The word "daily" has also been added to the figure caption to emphasise that daily values are the basis of the data. The caption now reads: "Figure 6: Climatic Variation on the Kamchatka Peninsula between 1948 and 2015, derived from NCEP/NCAR daily reanalysis figures averaged across the whole peninsula (see Kalnay at al., 1996). (A). Seasonal precipitation record. (B). Average seasonal temperature. (C). Summer temperature record from 1995-2015."

RC1.38

Comment: Figure 6b and d – what does glacier relief mean? Elevation? Elevation of what (min, mean, max)?

Response: The term 'relief' referred to 'altitudinal range'. We erroneously used both in the original manuscript.

Action: The word 'relief' has now been replaced with 'altitudinal range' throughout the text and in relevant figures. In fig 6 (now fig. 7), a graph showing the relationship with glacier length has now been added (this should have been included in the original manuscript).

Our response and associated actions relating to RC2's comments:

RC2.1

Comment: The paper is too long. The text contains lots of unnecessary naturally clear details and numbers which hinder its perception. Most numbers should be removed and put in the tables

Response: Despite the suggestion that the paper is 'too long', RC1 (point RC1.1) suggests that the paper is perhaps 'too concise'. The numbers mentioned in the text are also provided in tables. We feel that these numbers need to be discussed and noted in the text to facilitate description and discussion of the data.

Action: We have made no concerted effort to either lengthen or shorten the paper.

RC2.2

Comment: The authors don't seem to take into account the uniqueness of Kamchatka region as an area where Holocene volcanism exists along with glaciation and cannot help affecting it. The accomplished analysis is of a "standard" (formal) type, the same as for non-volcanic regions. However, along with tephra of recent eruptions (mentioned in the paper) many glaciers in Kamchatka (or their tongues) which motion is permanently destroying the friable slopes of Holocene volcanoes appear to be buried under a thick layer of deposits and continue flowing with it. At the same time they are nothing to do with rock glaciers having usually a snow covered accumulation zone and a high flow velocity. In other words, they keep all glacial features but have a thick cover of deposits on themselves which exceeds obviously 5cm, the threshold used in the paper for digitizing of glacial margins, which seems to be totally invalid for glacial mapping in Kamchatka. As a result the margins of most glaciers where the described effect is observed were digitized incorrectly and should be reconsidered by the authors. This conclusion is substantially based on the provided image of Tolbachik Volcano (Figure 2) where none of the glacial margins (neither conducted with semi-automated nor digitized manually) show the real extent of glaciers in the area (accepted and used by other authors, including USSR Glacier Inventory which data is used for comparison in the paper). This is a serious drawback which apparently can affect the results of the accomplished statistical analysis and the paper's conclusions in general. Now the conclusion that volcanic activity has no evident effect on glacier fluctuations seems to be very questionable and should be reconsidered using a new dataset with redigitized glacial margins.

Action: This issue relating to debris cover is addressed under our response to SC1.1. As noted in SC1.1, consideration of volcanic controls on glacier behaviour is now discussed in greater detail in section 5.3.4. (Re-named "Volcanic controls and debris cover").

RC2.3

Comment: Some glaciers appeared to increase their size. No possible explanations are presented.

Response: This apparent increase in size reflects small variations in glacier margins.

Action: This is addressed in our response to RC1.35

RC2.4

Comment: Kamchatka Peninsula is of particular interest because investigation of its recent glacial history has been limited (c.f., Khromova et al., 2014; Earl and Gardner, 2016) No overview or references to the glacial studies which were carried out. Mass-balance and paleoglaciological campaigns in Koryto glacier (Kronotskiy Peninsula); ice core boring of the crater glacier (Ushkovsky volcano); remote-sensing studies etc.

Action: This information has now been added to the manuscript (in section 2.3). This is addressed in detail in our response to RC1.8.

RC2.5

Comment: The peninsula has been shaped by both volcanic and glacial forces (Braitseva et al., 1995, 1997; Ponomareva et al., 2007, 2013; Barr and Clark, 2012 a,b) There are also folded mountains in Kamchatka by the way.

Response: We acknowledge this information, but are not convinced that its' directly relevant to the manuscript.

Action: No associated change has been made to the manuscript.

RC2.6

Comment: Subsequent estimates of the total number and surface area of glaciers on the peninsula tend to differ. For example, Muravyev (1999) reports 448 glaciers, with a total area of 906 km2, while Solomina et al. (2007) report 446 glaciers, with a total

area of 900 km2. But you have to know that for a half of the glaciation area penin-sula (Sredinny Range) data were provided from (Vinogradov, 1968) as new qualitative cartographic and satellite materials appeared on this area only in 21 centuries. At the same time for regions of an active volcanism the results leaning on the last decades of researches and for them for passed half a century any are given glacial retreating (especially "Rapid : : :") wasn't observed. Moreover, some glaciers in Klyuchevsky and Avacha groups of volcanoes continued the approach since last century, and, just, at the expense of the languages blocked by a moraine unaccounted by authors. Accord-ing to the latest published data on fluctuations of glaciers on the Kamchatka peninsula with use of aero-photo, good permission of space images, cartographic materials, and using of GPS receivers in field researches, really, some parts of modern glaciation retreating are observed. The truth this situation is peculiar mainly for glaciers of non-volcanic areas. Glaciers on volcanoes generally are in a steady state, or tend to growth. For example: – about glaciers of Koryaksky volcano – volcanic region (Manevich, Samoilenko, 2012): "Now there are seven glaciers with total area 8,36 km2. Three of them advance, two are in stationary state and one degrades; – glaciation of Sredinny Range (Muraviev, Nosenko, 2013): in the northern part of the Middle (Sredinny) Range from 1950s to 2002 – diminished by 16.6%; – glaciation of the not-volcanic regions (Muraviev, 2013): Glaciers on the Kronotsky Peninsula (Eastern Kamchatka) and the Alney-Chashakondzha massif (Sredinny Range) shrink. Since 1950 to 2013 area of Kronotsky glaciers reduced by 18.8 km2, or by 22.9% (for glaciers with areas larger 0.5 km2). Area of the Alney-Chashakondzha glaciers reduced for the period since 1950 to 2010 by 11.6 km2, or by 19.2%, that is comparable to similar characteristics of glacier systems of Altai, Tien Shan, and Caucasus, and this contraction correlates with changes of basic climatic variations, i.e. rising of summer air temperature and decreas-ing of solid precipitation. – the glaciers of Avachinskaya volcano group (Manevich et al., 2015): "The presentday glaciation of the Avachinskaya group was analyzed using data obtained during field works of 2007-2010. Twenty seven glaciers were founded the volcanic slopes with their total area of 24.043.6 km2. ... Comparison of recent data

with aerial photographs of 1974 has allowed estimating changes of the glacier front positions and to reconstruct the glacier dynamics for the last 40 years. Eighteen glaciers have been found to be in the stationary state, seven glaciers advance, and two glaciers degrade." – (Golub, Rassokhina, 2015): 1. Kropotkin's glacier (Eastern Kamchatka) since the end of the 1970th years of the XX century by 2014 (45 years) receded in the central part of the front in a place of its bifurcation on two tongues on 290 m, the left - on 390 m, the right - on 280 m. The area of a glacier was reduced by 34%; speed of retreating of Kropotkin glacier is increased, since 2003, that is caused by growth of summer air temperatures; thus, from 1976 to 2014 the area of glacier decreased from 0,85 sq.km to 0,29 sq.km, or for 34%, from which 2/3 of receding fall on the first 1.5 decades of 21 centuries. As far as I know, the russian glaciologists prepare publications on new inventory of glaciers of Klyuchevskaya group of volcanoes, etc., where the stable condition of glaciers of volcanic areas of the peninsula and retreating of glaciers non-volcanic areas similar to a velocity of retreating of other mountains glaciation of Eurasia are recorded.

Response: We are grateful to the reviewer for providing this information and associated referenced.

Action: Much of this information (and associated references) have now been added to section 2.3. See our response to RC1.8.

RC2.7

Comment: Glacier size (area, perimeter, length and relief): What is relief is totally unclear? There is no reference to the parameter is the part describing which glacial parameters were used in the statistical analysis.

Action: Please see our response to RC1.38, where we address this issue.

RC2.8

Comment: For each glacier identified in the inventory in 2000, maximum, minimum

Interactive
comment

and median altitude and mean surface slope were calculated from the SRTM DEM, and generalised glacier aspect was estimated from a line connecting the glacier's maximum and minimum altitudes. The mean annual receipt of solar radiation at the surface of each glacier was calculated using the Solar Radiation tool in ArcGIS (algorithms developed by Fu and Rich, 2002), and glacier length was estimated along inferred flowlines. Glaciers range in length from 170 m to 9930 m, while ranging in altitude from 273 m to 4407 m (a.s.l.), with a mean altitudinal range of 419 m, and a mean surface slope of 16.9. Rounded numbers should be provided.

Action: To address this, mean surface slope is now rounded up to $17°$.

RC2.8

Comment: The paper is to be shortened getting rid of unnecessary details describing the apparent parts of methodology. More stress should be put to the interpretation of the analysis results. Probably different glacial areas of peninsula can also be analysed against each other.

Action: Please see our response to RC2.1, where this is addressed. While comparing different parts of the peninsula might be interesting (and is work in progress), it would significantly lengthen this manuscript (by thousands of words), and as such we do not make regional comparisons here. We also feel that such information would detract from the paper's core message.

RC2.9

Comment: The statistical analysis should be done once again using the redigitized glacial margins, delineating the glacial parts covered by debris.

Action: This has now been done. Please see our response to SC1.1.

RC2.10

Comment: The conclusions should be tested with the new redigitized dataset. Con-
sidering the inevitable errors in such studies these two sources rather support than contradict other. The setting off is inappropriate here (Fig. 1).

Action: Again, this has now been addressed. Please see our response to SC1.1.

RC2.11

Comment: REFERENCES: Manevich T.M., Samoilenko S.B. Glaciers of the Koryak volcano // Ice and Snow. No.3 (119). 2012. 25-30 (Summary in English). Muraviev A.Ya. Nosenko G.A. Glaciation change in the northern part of the Middle Range on the Kamchatka Peninsula in the second half of XX century // Ice and Snow. No.2 (122). 2013. 5-11 (Summary in English). Muravyev A.Ya. Glacier size changes in Kronotsky Peninsula and Alney- Chashakondzha Massif, Kamchatka Peninsula in the second half of XX century and the beginning of XXI century // Ice and Snow. No.2 (126). 2014. 22-28 (Summary in English). Manevich T.M., Murav'ev Ya.D., Samoilenko S.B. Glaciers of the Avachinskaya volcano group: current condition (state) // Ice and Snow. Vol.55. No.3 (119). 2015. 14-26 (Summary in English). Golub N.V., Rassokhina L.I. The dynamics of the glaciers at Bolshoi Semyachik volcano (Eastern Kamchatka) at the beginning of the 21st century and formation of vegetation on the young moraines // Bull. Of Kamchatka regional association "Educational- Scientific Center". Earth Sciences. Vol.28, No.4. 2015. 60-71 (Abstract in English).

Response: We thank the reviewer for bringing these sources to our attention.

Action: All (with the exception of the final citation to Golub and Rassokhina, 2015) are now cited in section 2.3, and are included in the reference list.
* * *
Table. Comparison between statistics based on exposed glacier ice alone (original manuscript) and total glacier area (including debris covered areas) (revised manuscript).

|  | Exposed ice | Total glacier (including debris) |
|---|---|---|
| Number of glaciers (in 2000) | 676 | 673 |
| Number of glaciers (in 2014) | 766 | 738 |
| Total surface area (in 2000) km$^2$ | 664.8 ± 13.9 | 775.7 ± 27.9 |
| Total surface area (in 2014) km$^2$ | 465.1 ± 15.7 | 592.9 ± 20.4 |
| Area loss (2000 to 2014) km$^2$ (%) | 199.6 ± 7.2 km$^2$ (~ 30%) | 182.9 ± 6.6 km$^2$ (~ 24%) |
| Number of glaciers experiencing a reduction in area from 2000 to 2014 | 659 (~97%) | 654 (~97%) |
| Mean surface area in 2000 (Km$^2$) | 0.98 | 1.15 |
| Glacier length (min/max) (m) | 170/9930 | 170/20956 |
| Glacier altitude (min/max) (m.a.s.l.) | 273/4407 | 273/4407 |
| Mean altitudinal range (m) | 419 | 465 |
| Mean surface slope (°) | 16.9° | 17° |
| Mean annual receipt of solar radiation (min/mean/max) (kWh/m$^2$ ) | 57.5/111.9/169.5 | 489.8/817.1/1012.4 |

**Fig. 1.**

[Figure]

**Fig. 2.**

---

## Author Response (AR2)

**Responses and actions to comments**

**Comment: As also mentioned by the reviewer there is a substantial difference between your results and previous studies. This needs to be discussed more in detail. You should especially provide better information about the reasons for the difference to the Earl and Gardner inventory and show some overlays of outlines. This should be straight forward as the outlines are available.**

Action: To highlight the differences between the Earl and Gardner inventory and the inventory discussed by the paper a new figure has been generated (figure 8);

[Figure]

**Figure 8: Comparisons, from across Kamchatka, between our mapping (based on Landsat 8 imagery captured in 2014) and the mapping of Earl and Gardner (2016).**

Additionally this paragraph in section 5.1, has been reworded;

'In terms of glacier surface area, it is notable that our estimate from 2014 ($592.9 \pm 20.4$ km$^2$) differs significantly from Earl and Gardner's (2016) estimate (770.3 km$^2$). This we attribute to their semi-automated approach to mapping (i.e., NDSI), which can lead to an overestimation of glacier area as a result of snow patches being erroneously classified as glaciers (Man et al., 2014) (see Fig. 8), combined with the fact that their inventory was generated from a composite of satellite images, meaning their mapping does not reflect glacier extent during a single specified year (as noted in Sect. 2.3.).'

**Comment: P. 2 L. 7/8: It might be interesting to include the information (e.g. in brackets) for the readers who are not familiar with Russian that "sredinny" means "northern" and "vostocny" "eastern".**

Action: As suggested this has been changed and the sentence now reads;

'The peninsula has been shaped by both volcanic and glacial forces (Braitseva et al., 1995, 1997; Ponomareva et al., 2007, 2013; Barr and Clark, 2012 a, b), and is dominated by three mountain ranges: the Sredinny (Central) Range, the Eastern Volcanic Plateau and the Vostocny (Eastern) Range (Fig. 1). '

**Comment: P. 2 L. 29: Include the citation of the specific part/issue of the Soviet Glacier Inventory and provide more specific information about the actual years of the image sources which were used for glacier mapping.**

Action: More information has been added and now reads;

'At present, glaciers on the peninsula are found within the three principal mountain ranges, and on the high volcanic peaks of the Central Kamchatka Depression (see Fig. 1). The first attempts to estimate the extent of the region's glaciers were published as part of the 'Catalogue of Glaciers in the USSR, volume 20' (Vinogradov, 1968; Khromova et al., 2014). This was based on observations from topographic maps and aerial imagery captured in the early 1950s (various years), supplemented by field studies (Kotlyakov, 1980). In this survey, 405 glaciers were documented, with a total area of ~ 874 km$^2$ (Vinogradov, 1968).'

Additionally this reference has been added to the reference list:

Vinogradov, V.N.: Katalog lednikov SSSR, Kamchatka [Inventory of glaciers of the USSR, Kamchatka], v. 20: Leningrad, Gidrometeoizdat, 2–4, 1968. (In Russian)

**Comment: P. 3 L. 5/6: I do not agree that there is a real difference between the numbers presented by Muraviev and Solomina et al.. The numbers match actually quite well considering the uncertainty. Can you provide information from which the source of information. I assume that Solomina et al. (2007) did not map the glaciers by themselves.**

Response: This citation is included to demonstrate that various estimates are currently in use for the total number and extent of glaciers on the Kamchatka Peninsula. Solomina et al. (2007) do not provide a source for the estimates they quote.

Action: We have made no change.

**Comment: P. 3 L. 7: Any information about the size threshold Earl and Gardner used?**

Action: As suggested, additional information on the size thresholds Earl and Gardner used as part of their investigation has been added. This section now reads;

'Most recently, Earl and Gardner (2016) conducted extensive mapping of glaciers in Northern Asia using automated remote sensing techniques (i.e. NDSI), with a size threshold of 0.02 km$^2$, and documented 984 glaciers on Kamchatka, covering a total area of 770.3 km$^2$. This data was incorporated into the most recent version of the RGI (version 5, released in July 2015) (Arendt et al., 2015). '

**Comment: P. 4 L. 5/6: Include the information which processing level was used and where the data was downloaded. Level 1T scenes are orthorectified and usually no additional processing in necessary. If you did the orthorectification include the information how your performed it.**

Action: This sentence has been reworded and now reads;

'Orthorectified scenes were obtained from the USGS Earth Explorer website (http://earthexplorer.usgs.gov/) and projected using WGS 1984 Universal Transverse Mercator (UTM) Zone 57N, then filtered to find those with minimal cloud- and snow-cover.

**Comment: P. 4 L. 13: The scenes have usually a resolution of 30/15 m. But there are also scenes which were resampled to 28.5 m (e.g. if obtained from landcover.org). But then usually the panchromatic band was resampled to 14.25 m according to my knowledge. Please clarify.**

Action: The resolutions have been clarified and the sentence now reads;

'The Landsat scenes have a spatial resolution of 30 m for all bands except panchromatic band 8 (15 m) and Thermal Infrared Sensor (TIRS) band 6 (60 m), whilst the ASTER image (bands 1–3) has a spatial resolution of 15 m.'

**Comment: P. 4 L. 15 of "February" 2000, since the SRTM DEM was acquired 11-20 Feb. 2000. Include also which version of the SRTM DEM you used and where you downloaded the data.**

Action: As suggested this information is now included and the sentence now reads;

'To analyse glacier topography, SRTM 1 Arc-Second Global elevation data was obtained from the USGS Earth Explorer website. This reflects the surface topography of the glaciers in February 2000, with a horizontal resolution of ~ 30 m, and with vertical errors < 16m (at the 90% confidence level) (Farr et al., 2007).'

**Comment: P-5 L. 28: Not clear why only 15m. Did you map based on the panchromatic band or did you perform a resolution merge? Clarify in the methods section.**

Action: This section has now been reworded to clarify the method used and now reads;

Error was calculated following the method described in Bajracharya et al. (2014):

$$RMSE = \sqrt{\frac{\sum_{i=1}^{n}(a_i - \hat{a}_i)^2}{n}} \qquad (1)$$

where $a_i$ denotes glacier area and $\hat{a}_i$ is the glacier area calculated on the pixel base (i.e., the total number of pixels within a polygon, multiplied by the highest image resolution (15m) used for mapping), and n is the number of polygons digitised. Error was found to be ~ 3.6% in 2000 and ~ 3.4% in 2014. These values are comparable to those reported in other studies (Bolch et al., 2010; Bhambri et al., 2013; Bajracharya et al., 2014).

**Comment: P. 6 Chapter 4.1: Include here the information about the debris coverage (P. 7, 25ff)**

Action: As suggested this information has been moved with section 4.1 now reading;

'In total, 673 glaciers, with a combined surface area of 775.7 ± 27.9 km$^2$, were identified and mapped on the peninsula in 2000 (see Fig. 1). Summary statistics for these glaciers are presented in Table 2, revealing a mean surface area of 1.15 km$^2$ (ranging from 0.02 to 48.82 km$^2$), with ~ 77% < 1 km$^2$, and less than 4% > 5 km$^2$, though the latter represent just over one third of the total glacier area. Glaciers range in length from 0.17 km to 20.96 km, while ranging in altitude from 273 m to 4407 m (.a.s.l.), with a mean altitudinal range of 465 m, and a mean surface slope of 17°. The mean annual receipt of solar radiation at the surface of each glacier ranges from 489.8 to 1012.4 (kWh/m$^2$). Across the peninsula, 52 glaciers have tongues covered with debris (rock and ash), the majority of which exist on volcanic slopes in the Central Kamchatka Depression (see Yamaguchi., et al., 2007).

While this part of section 4.4 now reads;

'Based on mapping in 2000, ~ 8% (n = 52) of the peninsula's glaciers are classed as debris-covered. When the glacier population is split into 'debris-covered' and 'non debris-covered' samples in this way, the former lose on average ~ 11% (0.62 km$^2$) of their surface area between 2000 and 2014, while the latter lose ~ 44% (0.24 km$^2$). Thus, over the 2000–2014

period, debris-covered glaciers lose less surface area than non debris-covered examples, though this relationship is only statistically significant when values are expressed as a percentage, rather than in km$^2$.'

**Comment: P. 6 L. 14: I suggest to use km instead of m. for the glacier length.**

Action: As suggested, this has been changed, and the sentence now reads;

5 'Glaciers range in length from 0.17 km to 20.96 km, while ranging in altitude from 273 m to 4407 m (.a.s.l.), with a mean altitudinal range of 465 m, and a mean surface slope of 17°.'

Additionally, table 2 has been updated accordingly;

|  | Min | Mean | Max |
|---|---|---|---|
| Area (km$^2$) | 0.02 | 1.15 | 48.82 |
| Minimum Altitude (m.a.s.l) | 273 | 1277 | 2920 |
| Median altitude (m.a.s.l.) | 544 | 1506 | 3539 |
| Maximum altitude (m.a.s.l.) | 577 | 1742 | 4407 |
| Altitudinal range (m) | 17 | 465 | 3104 |
| Maximum flowline length (km) | 0.17 | 1.74 | 20.96 |
| Mean surface slope (°) | 5.7 | 17.0 | 35.5 |
| Mean annual solar radiation (kWh/m$^2$). | 489.8 | 817.1 | 1012.4 |

**Comment: P. 8: Section 5 Discussion- Compare your results not only to studies on Kamchatka but also to adjacent**
10 **glacierized mountain ranges to put your results into the regional (and maybe also into the global context). - A rough estimation of the glacier response time would be useful.**

Action: A new paragraph has been added to section 5.1 (found below), which has also been renamed as 'Comparison with other inventories'.

'Similar rates of retreat have been found for glaciers in other Asian mountain ranges. For example, in the Altai Mountains,
15 Narozhniy and Zemtsov (2011) document a 10.2% decrease in glacier area between 1956 and 2008, while Kamp and Pan (2015) document a 13% decrease between 1998/2001 and 2010/2011. In the Tien Shan Mountains, Farinotti et al. (2015) document an $18 \pm 6\%$ decrease in glacier surface area between 1961 and 2012. In the Kodar Mountains, Stokes et al. (2013) document a 44% decrease in the area of exposed glacial ice between 1963 and 2010, with a 40% loss since 1995, and in the Caucasus Mountains, Tielidze (2016) documents a $36.9 \pm 2.2\%$ decline in glacier surface area between 1960 and 2014,
20 though regional differences are noted. '

Additionally these references have been added to the references list;

Farinotti, D., Longuevergne, L., Moholdt, G., Duethmann, D., Mölg, T., Bolch, T., Vorogushyn, S. and Güntner, A.: Substantial glacier mass loss in the Tien Shan over the past 50 years, Nature Geoscience, 8, 716-723, 2015.

Fischer, M., Huss, M., Barboux, C. and Hoelzle, M.: The new Swiss Glacier Inventory SGI2010: relevance of using high-resolution source data in areas dominated by very small glaciers, Arctic, Antarctic, and Alpine Research, 46, 4, 933–945, 2014.

Narozhniy, Y. and Zemtsov, V.: Current state of the Altai glaciers (Russia) and trends over the period of instrumental observations 1952–2008, Ambio, 40, 6, 575-588, 2011.

Tielidze, L.G.: Glacier change over the last century, Caucasus Mountains, Georgia, observed from old topographical maps, Landsat and ASTER satellite imagery, The Cryosphere, 10, 713–725, 2016.

**Comment: P. 9 L. 5: The references are valuable but you need to consider that they address volume/mass changes and not area changes. Include also or replace by other relevant ones.**

Response: These references are here to merely highlight the fact that these are contributing factors to glacier fluctuations.

Action: we have made no change.

**Comment: P. 9 L. 9: Similar: Paul & Haeberli (2008) address volume changes.**

Response: These references were being used to highlight that smaller glaciers have a higher propensity for shrinking.

Action: We have replaced the reference, Paul & Haeberli (2008) with Racoviteanu et al (2015). This sentence now reads;

'Similar trends, with small glaciers showing a propensity to shrink rapidly, have been found in numerous regions globally (see Ramírez et al., 2001; Granshaw and Fountain, 2006; Racoviteanu et al., 2015; Tennant et al., 2012; Stokes et al., 2013).'

**Comment: P. 9 L: 14: Omit one "." after "warming.**

Action: As suggested this has been removed and the sentence now reads;

'This rapid decline in the area of smaller glaciers on the Kamchatka Peninsula could result in the loss of many over coming decades, as ~ 75% of the glaciers mapped in 2014 have an area < 0.5 km$^2$, of which ~ 87 % have a maximum altitude < 2000 m (a.s.l.), likely making them particularly sensitive to future warming.'

**Comment: Section: 5.3.2: Include and compare your results to similar other studies.**

Action: Glacier altitude. This section now reads;

'Comparisons between glacier median and maximum altitude and surface area fluctuations would appear to suggest that these factors exert some control on area change. However, the lack of any statistically significant relationship between area loss and minimum altitude might indicate that, rather than exerting a direct control on glacier area, glaciers with high maximum and median altitudes are typically the largest on the peninsula (i.e., there are positive and statistically significant relationships between glacier area and both maximum (r = 0.37) and median (r = 0.25) altitude), and that size exerts the primary control on glacier behaviour in this relationship. Similar trends, highlighting the relationship between glacier area and elevation, have been observed in other regions globally, for example in the Kanchenjunga–Sikkim region of the Himalayas (Racoviteanu et al., 2015) and in Cordillera Blanca, Peru (Racoviteanu et al., 2008).'

Additionally this reference has been added to the reference list:

Racoviteanu, A., Arnaud, Y., and Williams, M.: Decadal changes in glacier parameters in Cordillera Blanca, Peru derived from remote sensing, Journal of Glaciology, 54, 499–510, 2008.

**Comment: P. 10: Discuss the influence of debris-cover a bit more in depth, e.g. Vinogradov is a valuable reference, but there are many recent ones. Are there any glacial lakes/ice cliffs which may enhance the ice melt? You may also**
5 **consider that Pieczonka & Bolch found comparatively little area change (clearly below the global average) in Central Tien Shan (with many debris-covered glaciers), but the mass changes were similar to the global average. This might be due to the fact that debris-covered glaciers show less area changes than mass changes as also several other studies found in the Himalaya. I do not ask for an in depth analysis but 3-4 sentences in this regard would be very useful.**

Action: As recommended, further discussion was added on the influence of debris cover in section 5.3.4. This now reads;

[revised manuscript text omitted]